# Spectral efficiency and BER analysis of RNN based hybrid precoding for cell free massive MIMO under terahertz communication

Tadele A. Abose [1]*, Binyam G. Assefa[2], Yitbarek A. Mekonen[2], Naol W. Gudeta[2]

**1** Department of Electrical and Computer Engineering, Mattu University, Mattu, Ethiopia, **2** School of Electrical and Computer Engineering, Dire Dawa University, Dire Dawa, Ethiopia

* tadenegn@gmail.com

## Abstract

In future wireless networks, integrating Terahertz (THz) communication with cell-free massive multiple-input multiple-output (CFMM) systems presents a promising approach to achieving high data rates and low latency. This paper investigates the use of recurrent neural network (RNN)-based hybrid precoding in CFMM systems operating in the THz band. The proposed method jointly designs analog and digital precoders to adapt to dynamic channel conditions and user mobility. However, THz communication is challenged by high path loss and sparse scattering, which complicate accurate channel estimation. To address this, the RNN is trained to predict optimal precoding weights by learning spatial and temporal channel patterns, thereby improving channel estimation and mitigating pilot contamination. Simulation results show that the proposed method achieves higher spectral efficiency and lower bit error rate (BER) than conventional techniques. Specifically, the RNN-based approach attains a spectral efficiency of 10 bps/Hz at a signal-to-noise ratio (SNR) of 30 dB, compared to 8.2 bps/Hz for minimum mean square error (MMSE) precoding. For 16-QAM, the RNN-based method achieves a BER of $10^{-6}$ at an SNR of 11 dB, while MMSE requires 12.5 dB to reach the same BER. Overall, the RNN-based hybrid precoding consistently outperforms traditional methods across various SNR levels, antenna configurations, and user densities, underscoring its potential in next-generation THz wireless systems.

## 1. Introduction

Massive multiple input multiple output (mMIMO) is considered a revolutionary milestone in wireless communication systems, promising drastic improvements in spectral efficiency, network capacity, and signal quality. The main idea behind mMIMO is to deploy a vast number of antennas at both the base stations and user devices, unlike conventional MIMO systems, which use a limited number of antennas [1]. This

**Data availability statement:** All relevant data are within the manuscript and its Supporting Information files.

**Funding:** This work was supported by Mattu University (funding received by T.A.A.). The university provided limited support for software licensing. The University had no role in study design, data collection and analysis, decision to publish, or preparation of the manuscript.

**Competing interests:** The authors have declared that no competing interests exist.

significant increase in the number of antennas introduces a transformational dimension to wireless communication, enabling the simultaneous transmission of numerous data streams to many users [2]. This extraordinary level of spatial diversity is set to create game-changing capabilities by allowing better signal separation and boosting overall network performance [3].

With the exponential rise in data-driven applications and the emergence of technologies like the internet of things (IoT) and augmented reality (AR), the demand for high data rates and low latency in wireless communication networks has skyrocketed. To meet these needs, terahertz (THz) communication, operating in the frequency range of 0.1–10 THz, is viewed as a key technology for future wireless networks [4]. THz communication offers ultra-wide bandwidth, supporting data rates of several terabits per second, making it a strong contender for next-generation wireless systems [5].

THz communication presents unique challenges compared to mmWave due to its shorter wavelengths, which result in higher path loss and limited scattering capabilities. The high-frequency nature of THz waves leads to severe signal attenuation, requiring more precise beamforming techniques to maintain signal strength over longer distances. Additionally, hardware imperfections become more pronounced at THz frequencies, where components such as antennas and amplifiers must operate at extreme precision. These physical and technical challenges demand innovative solutions in system architecture, including high-dimensional signal processing, adaptive precoding, and advanced beamforming algorithms to ensure reliable and efficient communication [6].

mMIMO systems signify a paradigm shift from traditional cellular systems with centralized base stations. In contrast, CFMM systems deploy a large number of distributed access points (APs) that jointly serve users without predefined cell boundaries. This setup mitigates inter-cell interference and ensures consistent service quality, especially in high-density and mobile user environments [7].

To fully exploit CFMM in THz communication, hybrid precoding is essential. This approach combines digital and analog precoding to efficiently manage the vast number of antennas in mMIMO systems, balancing hardware complexity, power consumption, and performance. However, conventional hybrid precoding methods struggle with the rapidly changing channel conditions typical of THz frequencies [8].

Advancements in machine learning, particularly RNNs, provide promising solutions to these challenges. RNNs excel at handling sequential data and capturing temporal dependencies, making them ideal for predicting and adapting to varying channel conditions in real time. By integrating RNNs into the hybrid precoding process, we can dynamically adjust the precoding weights to optimize spectral efficiency and minimize the bit error rate (BER), even in the challenging THz band [9].

Numerous studies have highlighted the use of RNNs and other deep learning techniques to enhance mMIMO system performance [10]. RNN-based approaches have proven effective in tasks like channel estimation, beamforming, and interference mitigation, showing significant improvements over traditional methods. These advances

demonstrate the potential of RNNs to revolutionize hybrid precoding in CFMM systems, enabling effective utilization of THz communication capabilities [11].

This paper introduces a hybrid precoding scheme based on RNNs, specifically designed to address the dynamic channel conditions prevalent in THz communication. Our proposed scheme leverages the adaptive capabilities of RNNs to effectively manage precoding weights, optimizing performance in the rapidly fluctuating THz environment. Through a detailed analysis, we demonstrate that our RNN based hybrid precoding method significantly enhances spectral efficiency and reduces BER compared to traditional techniques. Moreover, our approach integrates advanced RNN components, such as long short-term memory (LSTM) networks, to further boost system performance in complex and dynamic wireless scenarios. To validate the efficacy of our scheme, we conduct extensive simulations within a CFMM framework, showcasing its superior performance over state-of-the-art systems. These contributions collectively underscore the potential of our method to advance THz communication capabilities, offering substantial improvements in both efficiency and reliability.

The main contributions of the paper are summarized as follows:

- Introduction of a Hybrid Precoding Scheme Based on RNNs: The paper presents a novel hybrid precoding scheme utilizing RNNs, specifically designed to address the dynamic channel conditions in THz communication. This scheme leverages the adaptive capabilities of RNNs to manage precoding weights effectively, optimizing performance in rapidly fluctuating THz environments.

- Enhanced Spectral Efficiency and Reduced BER: Detailed analysis demonstrates that the RNN-based hybrid precoding method significantly improves spectral efficiency and reduces BER compared to traditional techniques. The integration of advanced RNN components, such as LSTM networks, further boosts system performance in complex and dynamic wireless scenarios.

- Extensive Simulations and Validation: Extensive simulations within a CFMM framework validate the efficacy of the proposed scheme, showcasing its superior performance over state-of-the-art systems. These results underscore the potential of the method to advance THz communication capabilities, offering substantial improvements in both efficiency and reliability.

The structure of the paper is outlined as follows: Section 2 provides a comprehensive review of existing literature. Section 3 delves into the system model and problem formulation. Section 4 details the design of MMSE-based hybrid precoding for downlink multiuser massive MIMO (mMIMO) systems. Sections 5, 6, and 7 explore the pilot assignment algorithm aimed at mitigating pilot contamination in cell-free networks, the application of machine learning, and the presentation of results and discussions, including simulation outcomes, respectively. Finally, Section 8 wraps up with the paper's conclusions.

## 2. Literature review

The integration of RNNs into hybrid precoding schemes for cell-free massive MIMO systems operating under THz communication is an emerging research area that targets enhanced spectral efficiency and optimized bit error rate BER. Cell-free massive MIMO networks are characterized by their distributed nature, where multiple antennas serve users without the constraints of traditional cell boundaries. This allows for better connectivity and reduced interference, which is crucial in environments with high data demands such as those facilitated by THz frequencies [12].

The pursuit of enhancing spectral efficiency and reducing BER in wireless communication has driven extensive research, particularly in the development of CFMM systems integrated with THz communication [13]. As the demand for higher data rates and more efficient spectrum usage grows, CFMM systems have gained significant attention due to their ability to serve multiple users with minimal interference by utilizing distributed antenna arrays. These systems promise substantial gains in spectral efficiency and robustness, especially in densely populated environments. The integration of

THz communication, which operates at extremely high frequencies, further enhances the potential of CFMM systems by offering ultra-wide bandwidths capable of supporting the increasing demands of next-generation wireless networks, such as 6G and beyond. Additionally, researchers are exploring the use of machine learning and artificial intelligence to optimize system parameters in real-time, enhancing the adaptability of CFMM systems to dynamic wireless environments. Collectively, these cutting-edge approaches are paving the way for more reliable and efficient wireless communication systems, with CFMM and THz technologies at the forefront of this evolution [14].

Numerous studies have focused on evaluating the effectiveness of RNN-based hybrid precoding by comparing it to traditional methods and analyzing its performance across various channel conditions and system parameters. RNN-based hybrid precoding has shown promise in optimizing both spectral efficiency and energy efficiency in complex wireless communication environments, particularly in millimeter-wave (mmWave) and massive MIMO systems. Traditional precoding techniques often struggle to adapt to the rapidly changing channel conditions in these systems, leading to suboptimal performance. In contrast, RNN-based approaches can dynamically adjust the precoding process, leveraging their ability to model temporal dependencies and predict future channel states. Researchers have examined the performance of RNN-based precoding in scenarios with different levels of interference, mobility, and noise, demonstrating that it can significantly improve signal quality and reduce BER compared to conventional methods [15].

Hybrid precoding, which combines both analog and digital beamforming techniques, has been instrumental in improving the energy and spectral efficiencies of these systems. RNNs come into play by optimizing the beamforming process. Due to their capability of learning temporal dependencies and adapting to dynamic communication channels, RNNs help predict and adjust the beamforming parameters, significantly reducing errors in signal transmission and enhancing the system's overall performance, especially when deployed in high-frequency THz bands [16].

Recently, RNNs have emerged as effective tools for tackling the complexities of hybrid precoding in CFMM systems, especially under the demanding conditions of THz communication. The use of RNNs in this area is motivated by the need for low-complexity, yet efficient, precoding solutions that can handle the high-dimensional signal processing required at THz frequencies. Research indicates that RNN-based methods can dynamically adapt to varying channel conditions and maintain robust performance. Various studies have evaluated these techniques, focusing on their ability to optimize spectral efficiency and reduce BER across different scenarios [17].

Furthermore, spectral efficiency in cell-free massive MIMO systems can be substantially improved by employing hybrid precoding strategies that leverage RNNs for dynamic optimization. The distributed antenna systems in these networks contribute to a better signal-to-noise ratio (SNR) across users, which is crucial in THz communications. RNNs provide a predictive mechanism for adjusting precoding strategies in real-time, allowing for more efficient use of the available spectrum and enhancing the data throughput for multiple users in complex scenarios [18].

In their comprehensive survey, Zhang *et al*. [19] investigate the rapidly evolving domain of CFMM systems, underscoring their transformative potential in wireless communication by delivering reliable service and minimizing interference through the use of distributed antenna arrays. The paper delves into essential components of CFMM, including its fundamental principles, system architecture, and performance metrics. Notably, Zhang et al. point out the significant challenges and opportunities associated with the integration of CFMM and emerging technologies, such as THz communication. They emphasize the necessity for advanced signal processing methods, including hybrid precoding, to effectively manage the high-dimensional and intricate channel characteristics at THz frequencies. However, while the survey provides a valuable overview, it may benefit from a more detailed analysis of specific implementation strategies and the real-world implications of deploying CFMM in conjunction with THz technologies.

The authors explored the application of RNNs for adaptive hybrid precoding mmWave MIMO systems, a precursor technology to the more demanding THz communication. Their study emphasizes using RNNs to dynamically adjust hybrid precoding matrices, effectively balancing digital and analog components to enhance signal transmission efficiency. This adaptive method is particularly advantageous in mmWave systems, where rapid channel fluctuations and

high-dimensional signal processing demands are prevalent [20]. However, while their research provides a critical foundation for employing RNNs in high-frequency communication systems, it mainly focuses on mmWave scenarios. To fully extend these advancements to CFMM systems under THz communication, further research is necessary to address the unique challenges posed by THz frequencies.

The authors [21] investigate the effectiveness of RNN-based hybrid precoding in mMIMO systems, particularly in situations where hardware imperfections can significantly affect system performance. This research is essential for comprehending the practical implementation of CFMM systems in THz communication, where similar issues are anticipated. RNNs contribute to maintaining high spectral efficiency and low Bit Error Rate (BER). However, the study predominantly focuses on mmWave frequencies, leaving a gap in understanding how these findings apply to the THz band, where the impact of hardware imperfections is likely to be more severe. Moreover, although the RNN-based approach exhibits resilience to hardware flaws, the challenges related to its computational complexity and real-time processing capabilities in large-scale CFMM systems require further optimization and exploration to ensure practical applicability.

Sun, Yu, Hu, and Peng [22] investigate the application of RNN-based precoding for mMIMO systems, focusing on enhancing spectral efficiency and reducing BER. Their work leverages the dynamic learning capabilities of RNNs to optimize precoding matrices in response to real-time channel variations. The authors [20] demonstrate that RNN-based precoding can significantly outperform traditional linear and nonlinear precoding methods by efficiently managing the high-dimensional signal processing requirements inherent in mMIMO systems. However, their study is primarily focused on conventional mMIMO setups and does not fully address the unique challenges posed by CFMM systems under THz communication.

The use of RNNs in hybrid precoding also aids in reducing the BER by effectively mitigating the effects of interference and signal distortion, which are prominent at THz frequencies. Traditional linear precoding techniques often struggle in such high-bandwidth systems due to the nonlinear nature of the channels, especially in dynamic or dense user environments. By applying machine learning techniques, including RNNs, the precoding process can dynamically adapt to the channel variations, improving both the reliability and the efficiency of data transmission [23].

In summary, while extensive research has demonstrated the potential of RNN-based hybrid precoding to enhance spectral efficiency and reduce BER in mmWave MIMO systems, significant gaps remain in the context of CFMM systems operating under THz communication. Existing studies primarily focus on mmWave frequencies and conventional MIMO setups, often overlooking the unique challenges posed by THz frequencies. Moreover, while RNNs have shown resilience against hardware imperfections and the ability to dynamically adapt to varying channel conditions, their computational complexity and real-time processing capabilities in large-scale CFMM systems require further optimization. Our work addresses these gaps by proposing a novel RNN-based hybrid precoding method specifically tailored for CFMM systems in the THz band. This approach aims to dynamically optimize precoding weights to improve spectral efficiency and minimize BER, overcoming the limitations of traditional methods and paving the way for the practical deployment of next-generation wireless networks.

Expanding on insights from the literature review on RNN-based hybrid precoding methods in wireless communications, we now transition to Section 3. In this section, we outline the system model and problem formulation for our proposed hybrid precoding scheme, which is tailored for CFMM systems operating at THz frequencies.

## 3. System model and problem formulation

This paper examines a multiuser CFMM system that employs a fully connected hybrid precoding structure. Unlike conventional cellular networks, this system operates within a cell-free architecture, where traditional cell boundaries are removed. By eliminating these boundaries, the network distributes numerous access points (APs) across the entire coverage area. This decentralized deployment of APs improves connectivity, allowing multiple users to be served simultaneously, without the limitations imposed by predefined cell regions. The distributed nature of the APs optimizes resource allocation and enhances the overall system performance, as depicted in Fig 1.

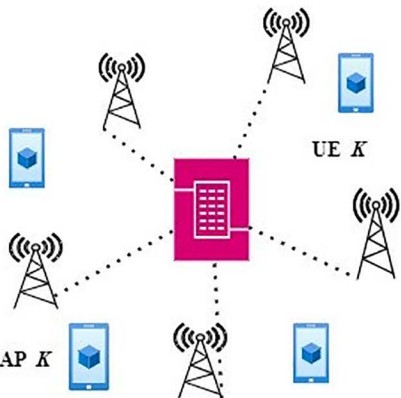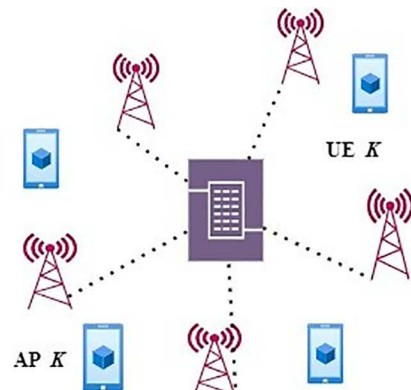

**Fig 1. CFMM architecture.**

Each AP has $N_{TR}$ antennas and $K$ RF chains to serve $K$ user equipment (UE). The statement "Each AP serves $K$ UEs" refers to a system model where the total number of UEs in the simulation is fixed to $K$ per AP. That is, we modeled the system with a limited number of $K$ UEs per AP for simplicity and scalability in simulation. AP selection or dynamic user association was not applied On the UE side, each user is equipped with $N_{RT}$ antennas and an RF chain. Both sides of Fig 1 represent different system configurations with the same set of UEs and APs. The labels are intentionally kept identical to emphasize that the comparison is based on network architecture, not user or access point identity. In CFMM [24], where the traditional cell boundaries are dissolved, each access point (AP) applies a digital baseband precoder of size $K \times K$, represented by $\boldsymbol{V_B} = [\boldsymbol{v}_{B_1}, \ldots \boldsymbol{v}_{B_k}]$, followed by an analog precoder $V_R$. The analog precoding matrix $V_R$ has a dimension of $N_{TR} \times K$ and is defined as follows

$$V_R = [v_{R_1}, \ldots \ldots v_{R_k}] \tag{1}$$

where, $\boldsymbol{V_R}$ are normalized to satisfy the constant modulus constraints $|(V_R)_{i,j}| = \frac{1}{\sqrt{N_{TR}}}, \forall i, j$

Let $s_k$ denote the transmitted signal at the AP for the $kth$UE and $\boldsymbol{s} = [s_1, \ldots \ldots, s_k]^T \in C^{k \times 1}$, such that the transmitted signal by the APs can be given by

$$\boldsymbol{x} = \sum_{k=1}^{K} \boldsymbol{v_k} s_k = \boldsymbol{V_s} \tag{2}$$

where, $\boldsymbol{v_k} = \boldsymbol{V_R} \boldsymbol{v}_{B_k}$, $\boldsymbol{v}_{B_k} \in C^{k \times 1}$ is digital precoding vector for the $kth$ user and $\boldsymbol{V} = [\boldsymbol{v}_1, \ldots \ldots \boldsymbol{v}_k] \in C^{N_{TR} \times K}$ are complete hybrid precoding metrics. Then the received signal $\boldsymbol{y}_k$ at the $kth$ user terminal is

$$\boldsymbol{y_k} = \boldsymbol{H_k} \boldsymbol{v_k} s_k + \boldsymbol{H_k} \sum_{k \neq j} \boldsymbol{v_j} s_j + \boldsymbol{n_k} \tag{3}$$

- $\boldsymbol{H_k} \in C^{N_{TR} \times K}$ The channel matrix between the transmitter and the $kth$ user terminal.

- $s_k$ Transmitted signal for the $kth$ user.

- $\boldsymbol{n_k}$ Additive Gaussian noise at the $kth$ user terminal.

In this study, frequency offset was not explicitly considered in Equation (3) as the focus is on evaluating the performance of the proposed hybrid precoding method under idealized conditions. The omission of frequency offset allows for isolating the effects of the precoding scheme and channel characteristics without introducing additional complexity.

The received signal can be formed after combining at each user terminal is given as

$$z_k = W_{R_k}{}^H H_k v_k s_k + W_{R_k}{}^H H_k \sum_{k \neq j} v_j s_j + W_{R_k}{}^H n_k \tag{4}$$

where, $(W_{R_k})^H \in C^{1 \times N_{TR}}$ is analog combiner at the *kth* user terminal. $(\cdot)^H$ denotes the Hermitian transpose (conjugate transpose) of a complex-valued matrix or vector and $(\cdot)^T$ denotes the transpose of a matrix or vector. $H_k$ of dimension $N_{RT} \times N_{RT}$ represent the THz channel from APs to the *kth* user and $n_k$ is the noise at the receiver which is assumed to fellow Gaussian distributed with zero mean and variance $\sigma^2 I$. The effective channel (antenna array gain) of the *kth* user is

$$h_{l_k} = W_{R_k}{}^H H_k V_R \tag{5}$$

In the context of CFMM, the signal-to-interference-plus-noise ratio (SINR) at the kth user terminal is a crucial metric for assessing system performance [25].

$$\text{SINR}_k = \frac{\frac{p}{K} \left| h_{e_k} v_{B_k} \right|^2}{\sum_{k \neq j}^{K} \cdot \frac{p}{K} \left| h_{e_k} v_{B_j} \right|^2 + \sigma^2} \tag{6}$$

Achievable rate of the *kth* user terminal (UE) represents the achievable data rate and is computed using the SINR. It is given

$$r_k = \log\left(1 + SINR_k\right)$$

The spectral efficiency in a cell-free mMIMO system can be calculated based on the Shannon capacity formula, which takes into account the achievable data rate $r_k$ and the bandwidth ($B$)

$$SE_k = \frac{r_k}{B} \tag{8}$$

where B denotes the system bandwidth used to normalize the achievable rate $r_k$ into spectral efficiency units of bits/sec/Hz.

### 3.1 Channel model

Due to the short wavelength of THz signals, their transmission suffers from significant attenuation and minimal scattering effects, leading to predominantly line-of-sight (LoS) propagation. Challenges in THz communication include propagation losses, limited scattering, hardware constraints, thermal noise, and atmospheric absorption. Advanced machine learning methods perform channel estimation in THz communication. The channel is a fading channel model tailored for THz communications. It incorporates scattering clusters and propagation paths ($Nc$ and $Np$) with random gains, azimuth, and elevation angles, capturing spatial and angular diversity. The model accounts for antenna array response at the transmitter and receiver while considering atmospheric effects typical of THz environments. Atmospheric turbulence can significantly impact THz communication channels, as variations in temperature and pressure cause fluctuations in the refractive index, leading to phase and amplitude distortions in the transmitted signals. These fluctuations introduce random fading effects in the channel, thereby degrading signal quality and increasing bit error rates. In particular, the high sensitivity of THz frequencies to turbulence amplifies issues like scintillation, which further complicates channel estimation and correction methods in THz CFMM systems. [26,27].

In this paper, a geometric channel model with scatterers is employed to characterize the THz channel for each user. This approach is fundamentally different from a Rayleigh fading model. While Rayleigh fading assumes a rich scattering

environment without a line-of-sight (LoS) component and models channel gains as complex Gaussian random variables with zero mean (i.e., $CN(0,1)$. our geometric channel model explicitly incorporates a finite number of propagation paths (Np), each with specific gains, angles of arrival and departure (azimuth and elevation), and scatterer positions. This allows us to capture the LoS and sparse multipath nature of THz propagation more realistically. The mention of $CN(0,1)$. in Eq. (20) and Section 7 was intended only for simplifying the small-scale fading component in some analytical contexts and does not reflect the full model used in the simulations. normalized antenna response vectors. The mean cluster angles are uniformly distributed over [0,2π] reflecting the spatial randomness typical of high-frequency THz channels. It is well-suited for THz CFMM systems, where the propagation environment is dominated by a limited number of significant scatterers, and the impact of beam misalignment, phase front warping, and atmospheric effects is pronounced [28]. Thus, the channel matrix for the *kth* user on the *kth* subcarrier in the THz CFMM systems can be represented as follows:

$$H_k = \sqrt{\frac{N_{TR}N_{RT}}{N_C N_P}} \sum_{i=0}^{Nc} \sum_{i=0}^{Np} a_{ik,il} \mathbf{a}_r(\varnothing_{k,il}^r, \theta_{k,il}^r) \mathbf{a}_r(\varnothing_{k,il}^t, \theta_{k,il}^t)$$

(9)

where, $N_c$ is the number of scattering clusters and each cluster contributes $N_p$ paths. $a_{ik,il}$ is the gain of the *ith* ray in the *ith* scattering cluster. $\mathbf{a}_r(\varnothing_{k,il}^r, \theta_{k,il}^r)$ and $\mathbf{a}_r(\varnothing_{k,il}^t, \theta_{k,il}^t)$ represent the normalized antenna array response vectors at the receiver and transmitter, respectively. $\varnothing_{k,il}^r(\theta_{k,il}^r)$ and $\varnothing_{k,il}^t(\theta_{k,il}^t)$ are the azimuth (elevation) angles of the *kth* ray in the *ith* scattering cluster on the receiving and transmitting sides, respectively. The mean cluster angles $\varnothing_{k,il}^r(\theta_{k,il}^r)$ and $\varnothing_{k,il}^t(\theta_{k,il}^t)$ are subject to uniform random distribution in the range of [0,2π].

The channel model in this paper utilizes a geometric fading model to better represent the propagation characteristics of THz communication. This model accounts for scattering and LoS propagation typical in THz environments, incorporating scattering clusters and individual propagation paths (5 paths $N_P$ and 10 clusters $N_C$), each with random gains and azimuth/elevation angles. As explained in Section 7, The channel matrix is derived by considering the normalized antenna array responses at both the transmitter and receiver, as well as the impact of atmospheric effects such as turbulence, which can lead to phase and amplitude distortions. By leveraging advanced machine learning methods for precoding weight estimation, this approach effectively addresses the challenges posed by atmospheric absorption, thermal noise, and the minimal scattering of THz signals. The dataset used for training includes real and imaginary channel gains, channel phases, spatial correlation between antennas, noise power, transmitted signal power, path loss, shadowing, and fading effects, which are captured in signal strength, phase shift, spatial correlation between antenna elements, noise, signal attenuation, and environmental fluctuations. This enables more accurate modeling of the THz communication channel for CFMM systems.

The primary challenge in using Low-Density Parity-Check (LDPC) codes for THz communication systems, particularly in a CFMM context, is that while LDPC improves error correction, its contribution to overall system performance might be limited compared to the dynamic channel adaptation capabilities of advanced precoding techniques like the RNN-based hybrid precoding we propose. THz communication channels suffer from high path loss, limited scattering, and severe attenuation, which are more effectively addressed by optimizing the channel estimation and precoding strategies rather than solely relying on error correction techniques. although LDPC and other error correction methods are crucial for ensuring data integrity, their performance gains tend to saturate as channel impairments become more severe at higher frequencies, such as in the THz band. Studies have shown that under such conditions, improving channel estimation and beamforming can result in greater spectral efficiency and lower bit error rates compared to relying primarily on error correction techniques [29]

### 3.2 Problem formulation

In multiuser mMIMO systems, hybrid precoding is primarily formulated as an optimization problem, with the main goal of maximizing the overall sum-rate ($r_k$) for all users, as referenced in [30]. The challenge lies in determining the most

effective balance between digital and analog precoding techniques, ensuring optimal signal transmission and improving data throughput across the network. By optimizing this balance, the system seeks to enhance the efficiency of resource utilization, resulting in better performance for each user while maintaining high overall network capacity and minimizing interference. This approach is crucial for maximizing the spectral efficiency and achieving robust communication in multiuser scenarios.

$$\max_{\boldsymbol{V}_R \boldsymbol{W}_{R_k} v_{B_k}} \sum_{k=1}^{K} r_k \tag{10}$$

$$s.t \quad C_1 : \quad |(\boldsymbol{V}_R)_{i,j}| = \frac{1}{\sqrt{N_{TR}}}, \forall i,j$$

$$C_2 : \quad |(\boldsymbol{W}_{R_k})_{i,j}| = \frac{1}{\sqrt{N_{TR}}}, \forall i,k$$

$$C_3 : \quad \|\boldsymbol{V}\|_F^2 = K$$

Deriving the nonconvex optimization directly does not give rise to a straightforward solution [31,32]. However, this can be decomposed into analog and digital parts. For the maximization of the antenna array gain, the effective channel $\boldsymbol{h}_{l_k} = \boldsymbol{W}_{R_k}{}^H \boldsymbol{H}_k \boldsymbol{V}_R$ provides a solution in the multiuser mMIMO system. It should be noted that this formulation does not include a mechanism for per-user quality-of-service (QoS) equalization. As a result, the observed system behavior may favor users in better channel conditions, potentially overlooking fairness among UEs. This design choice was intentional to focus on overall spectral efficiency improvements, and future work will incorporate fairness-aware formulations.

$$\max_{\boldsymbol{V}_R \boldsymbol{W}_{R_k}} \sum_{k=1}^{K} \left\| \boldsymbol{W}_{R_k}{}^H \boldsymbol{H}_k \boldsymbol{V}_R \right\|_F^2 \tag{11}$$

$$s.t. \quad C_1 : \quad |(\boldsymbol{V}_R)_{i,j}| = \frac{1}{\sqrt{N_{TR}}}, \forall i,j$$

$$C_2 : \quad |(\boldsymbol{W}_{R_k})_{i,j}| = \frac{1}{\sqrt{N_{TR}}}, \forall i,k$$

The digital optimization problem can be restated

$$\max_{v_{B_k}} \quad \sum_{k=1}^{K} r_k \tag{12}$$

$$s.t. \quad C_1 : \quad \|\boldsymbol{V}\|_F^2 = K$$

In Section 3, we established the system model for our CFMM framework and formulated the problem of optimizing hybrid precoding for THz communication. Building on this foundation, Section 4 introduces a Minimum Mean Square Error (MMSE)-based hybrid precoding design for the downlink multi-user mMIMO system. This approach aims to improve our precoding scheme's performance by addressing identified challenges.

## 4. The MMSE based hybrid precoding design for downlink MU mMIMO system

### 4.1 Analog precoder and combiner design

MMSE criterion is employed to minimize the mean square error (MSE) of the data streams sent to the UE. The expected value of the squared norm of the difference between the transmitted and received signals, $\mathbb{E}[\| s - z \|^2]$, is minimized. This approach helps to derive the closed-form MMSE-based analog precoder for the *kth* UE under the assumption of full channel state information (CSI) availability [33].

$$A_k = (H_k^H H_k + \frac{K\sigma^2}{P}I)^{-1} H_k^H \tag{13}$$

where, $K$ is the number of users, $P$ is the transmit power, $\sigma^2$ *is* the noise and is the $I$ is the identity matrix variance the MMSE based analog precoder relies entirely on the channel knowledge, denoted as $H_k$. the MMSE based approach can be used to determine the analog precoder effectively.

$$v_{R_k}(i,j) = \frac{1}{\sqrt{N_{TR}}} \exp(j\theta_{i,j}) \tag{14}$$

where, $\theta_{i,j}$ is the angle of the $(i,j)th$ element of matrix $A_k$. Given the analog precoder $V_R$ and the channel $H_k$, let

$$B_K = H_k V_R \tag{15}$$

The RF combiner can also be designed based on the MMSE criterion as follows

$$G_k = ((B_K)^H B_K + \frac{K\sigma^2}{P}I)^{-1} (B_K)^H \tag{16}$$

The analog combiner for the fully-connected one $w_{R_k}$ can be stated as follows

$$w_{R_k}(i,j) = \frac{1}{\sqrt{N_{RT}}} \exp(j\psi_{i,j}) \tag{17}$$

where, $\psi_{i,j}$ is the phase angle of the $(i,j)th$ element of $G_k$

### 4.2 Digital precoder design

The design of the analog precoder in multiuser mMIMO systems is aimed at leveraging the passive gain provided by the antenna array. By utilizing the analog precoder, denoted as $V_R$, in combination with the channel matrix $H_k$ and the RF combiner formed from the vectors $[w_{R_1}, \ldots, w_{R_k}]$, an effective digital precoding scheme can be developed to manage and mitigate user interference. The analog precoder works to maximize signal strength by aligning the signals with the array's physical structure, while the digital precoder further refines the transmission to ensure that interference between users is minimized. In this context, a Minimum Mean Square Error (MMSE)-based digital precoder is applied, offering a precise way to control and reduce inter-user interference, thus optimizing overall system performance and improving data transmission accuracy.

$$V_B = \left( H_e^H H_e + \frac{K\sigma^2}{P}I \right)^{-1} H_e^H \tag{18}$$

The power constraints can be satisfied by normalizing the digital precoder $V_B$ for the kth user terminal as follows

$$\widetilde{v}_{B_k} = \frac{v_{B_k}\sqrt{K}}{\| V_R v_{B_k} \|_F} \tag{19}$$

```
Algorithm 1. MMSE based hybrid precoding
Require: H_k and P
For k = 1, K Do
        A_k = (H_k^H H_k + (Kσ²/P)I)^{-1} H_k^H
    Find the Analog Precoder  v_{R_k}(i,j) = (1/√N_{TR}) exp(jθ_{i,j})
        B_K = H_k V_R
G_k = ((B_K)^H B_K + (Kσ²/P)I)^{-1}(B_K)^H
w_{R_k}(i,j) = (1/√N_{RT}) exp(jψ_{i,j})
End for
    Calculate effective channel  h_{e_k} = v_{R_k}^H H_k^H W_R
Digital Part
    Find the Digital Precoder  V_B = (H_e^H H_e + (Kσ²/P)I)^{-1} H_e^H
    Normalize the Digital Precoder  ṽ_{B_k} = v_{B_k}√K / ||V_R v_{B_k}||_F
```

In Section 4, we presented an MMSE-based hybrid precoding design for the downlink multi-user CFMM system. With this framework in place, we now move to Section 5 to address pilot contamination in cell-free networks.

## 5. Pilot assignment algorithm to mitigate pilot contamination for cell free networks

In this section, we explore the CFMM system, which comprises M access points (APs) each with N antennas, and $K$ user equipment's (UEs) with multiple antennas, randomly distributed across a coverage area. Though typically $M \gg K$, CFMM systems perform efficiently regardless of these values due to their scalable architecture. Each AP is connected to a central processing unit (CPU) via a backhaul network. the communication channel between the *k-th* UE and the *m-th* AP is modeled as [34,35]

$$h_{mk} \sim NC(0, R_{mk}) \tag{20}$$

where, $R_{mk}$ is the spatial correlation matrix incorporating shadowing and path loss features, with dimensions $R_{mk} \in C^{N \times N}$. The large scale fading coefficient $\beta_{mk}$, is derived from the trace of $R_{mk}$, given by $\beta_{mk} \triangleq \frac{t_{r(R_{mk})}^N}{N}$, CFMM systems operate in TDD mode, with the total time $\tau_c$ in a coherence block split into pilot transmission $\tau_p$, uplink (UL) data transmission $\tau_{UL}$, and downlink (DL) data transmission $\tau_{DL}$.

### 5.1 Pilot contamination (PC) mitigation

Pilot contamination occurs when different UEs use the same pilot sequences, leading to interference [36]. To combat this, an efficient pilot assignment strategy is utilized. Each UE is allocated a pilot based on its distance to the nearest AP or, if multiple APs are equidistant, based on the large-scale fading coefficient $\beta$ [37].
    Assignment Algorithm

- **UE Selection:** Prioritize UEs with poor channel quality for pilot assignment. Calculate and sort the large-scale fading coefficients $\beta$ for all UEs in descending order.

- **AP Selection:** UEs use synchronization signals to communicate with nearby APs. The closest AP serves the UE. If multiple APs are equidistant, the AP with the highest $\beta$ is chosen.

- **Pilot Allocation:** The serving AP assigns an unused pilot to the UE and informs neighboring APs to avoid using the same pilot, thereby reducing PC and enhancing system performance.

### 5.2 Received pilot signal

When UEs transmit pilot signals, the signal received by the m-th AP is:

$$Z_m^p = \sum_{i=1}^k \sqrt{\rho_i \tau_p} h_{im} \varnothing_k + n_m^p \tag{21}$$

where, $n_m^p$ is the noise power, $\rho_i$ is the $i^{th}$ UE's UL pilot power, and $n_m^p \in C^{N \times \tau_p}$ is the received noise matrix, or $n_m^p \sim NC(0, \sigma^2)$

### 5.3 Pilot contamination (PC) and system performance

In scenarios where $\mathbf{K} \leq \tau_p$, pilot sequences are shared among UEs, leading to PC. This interference reduces system performance and degrades the quality of channel estimates. The relationship between PC and the number of AP antennas $N$ is critical; higher $N$ can mitigate the negative impact of PC, but careful pilot allocation is essential to maintain system performance.

### 5.4 Optimal pilot allocation algorithm

The algorithm for optimal pilot allocation aims to minimize PC and optimize the quality of channel estimates:

- **Initialization**: Start with the set of UEs $U_A = U_L$,

- **Beta Calculation**: Compute $\beta$ for all UEs and sort them.

- **Pilot Assignment**:

  - **if $K \leq \tau_p$ assign** $\varnothing_k$ to the $k^{th}$ user

  - **if $\mathbf{K} \geq \tau_p$** find the best serving AP for each UE and allocate the pilot that minimizes interference.

The paper utilizes a pilot allocation algorithm that provides robust solutions for enhancing the quality of channel estimation and overall system performance. This algorithm intelligently assigns pilots based on large-scale fading coefficients and the spatial distribution of UEs relative to APs, thus minimizing interference and optimizing resource utilization. Additionally, the systematic UE selection process prioritizes those with poor channel quality, ensuring efficient use of resources. These innovative strategies make this approach highly relevant and effective for improving communication in CFMM systems. In Section 5, we tackled pilot contamination in cell-free networks with an innovative pilot assignment algorithm. With pilot contamination mitigated, Section 6 focuses on integrating machine learning techniques to boost CFMM system performance.

## 6 Machine learning

Machine Learning (ML) and Deep Learning (DL) have transformed wireless communication by enabling adaptive, data-driven system improvements [38]. In THz communication, where estimating channel parameters is complex, neural networks like Convolutional Neural Networks (CNNs) and RNNs excel in predicting Channel State Information (CSI). CNNs can be effective for spatial feature extraction in channel estimation, while reinforcement learning could enhance adaptive beamforming by optimizing decisions based on real-time feedback, further improving performance in THz communication. [39]. RNNs, particularly LSTM networks, handle sequential data effectively, modeling dynamic behaviors crucial for THz frequencies. These advanced architectures enhance spectral efficiency and reduce BER in next generation wireless networks [40].

### 6.1 LSTM network architecture for hybrid precoding

LSTM networks are well suited for tasks requiring sequential and temporal data dependencies. In hybrid precoding for CFMM systems under THz communication, LSTMs play a crucial role by dynamically learning and adapting precoding weights based on evolving channel conditions. The LSTM network architecture involves multiple layers to capture complex patterns in CSI. CSI refers to the information that describes the properties of the communication channel, such as signal attenuation and phase shift, which is crucial for adapting the transmission strategy to optimize

performance. On the other hand, coding diversity involves using various coding techniques to mitigate the adverse effects of fading and interference, ensuring more reliable data transmission. Integrating these concepts enhances the system's ability to adapt its beamforming strategies, ultimately improving spectral efficiency and communication quality. The input layer processes CSI data, and subsequent LSTM layers maintain and update the network's internal state, capturing short-term fluctuations and long-term trends in channel conditions [41]. This temporal processing capability is vital for THz communication, where channel conditions change rapidly and require real-time adaptation. The output layer generates optimized weights for analog and digital precoding stages, enhancing spectral efficiency and minimizing BER [42].

## 6.2 RNN based hybrid precoding

This adaptation makes the hybrid precoder system capable of optimizing beamforming strategies based on the information learned from RNN training outputs. RNNs dynamically adjust both analog and digital precoders based on changing channel conditions. In hybrid precoding, the goal is to optimize beamforming by selecting the best weights for the analog and digital precoders to maximize spectral efficiency and minimize BER. This integration allows the system to adjust its beamforming parameters over time according to changes in channel conditions and user requirements, thereby enhancing overall performance, spectral efficiency, and communication quality. In contrast to other types of neural networks, such as Convolutional Neural Networks (CNNs), which are particularly effective at extracting spatial features, Recurrent Neural Networks (RNNs) offer a unique advantage in handling temporal dynamics. RNNs are specifically designed to capture sequential dependencies, making them highly adaptable to time-varying data. This temporal adaptability is especially valuable in wireless communication systems, as it allows RNNs to dynamically update precoding parameters in response to short-term fluctuations in the communication channel. This capability becomes critical in high-mobility scenarios, such as THz communication, where rapid changes in the channel conditions must be continuously tracked and managed to ensure reliable and efficient data transmission, as highlighted in [43].

Fig 2 represents the hybrid precoding structure enabled by RNN in CFMM under THz communication. An RNN is used to handle beamforming and input from the channel matrix, spatial correlation, and power fading to optimize the beamforming process.

Workflow

- **Initialization:** The dynamic channel data and precoding information are supplied to the RNN.

- **Precoder Generation:** The RNN generates analog and digital precoder and analog combiner parameters to enhance system performance.

- **Hybrid Precoder Update:** The RNN outputs update on hybrid digital and analog precoders.

- **Transmission and Reception:** The transmitted signal will be processed by the hybrid precoder parameters, which at the receiver side should be processed with the RNN predicted analog combiner parameters.

Main data generation parameters for the RNN-based hybrid precoding in CFMM under THz include real and imaginary channel gains, channel phases, spatial correlation between antennas, noise power, transmitted signal power, path loss, shadowing, and fading effects. These are captured in signal strength, phase shift, correlation, noise, signal attenuation, and environmental fluctuations [44]. Besides, the precoding weights of digital and analog parts, as well as the combining matrix of the analog part, directly control the hybrid precoding process and make the data set under the hybrid precoding link extremely diverse, which is crucial for training and testing the RNN model in a robust THz communication environment. This integration contributes to enabling the system to adapt its beamforming strategies toward achieving the goal of better performance in terms of spectral efficiency and communication quality.

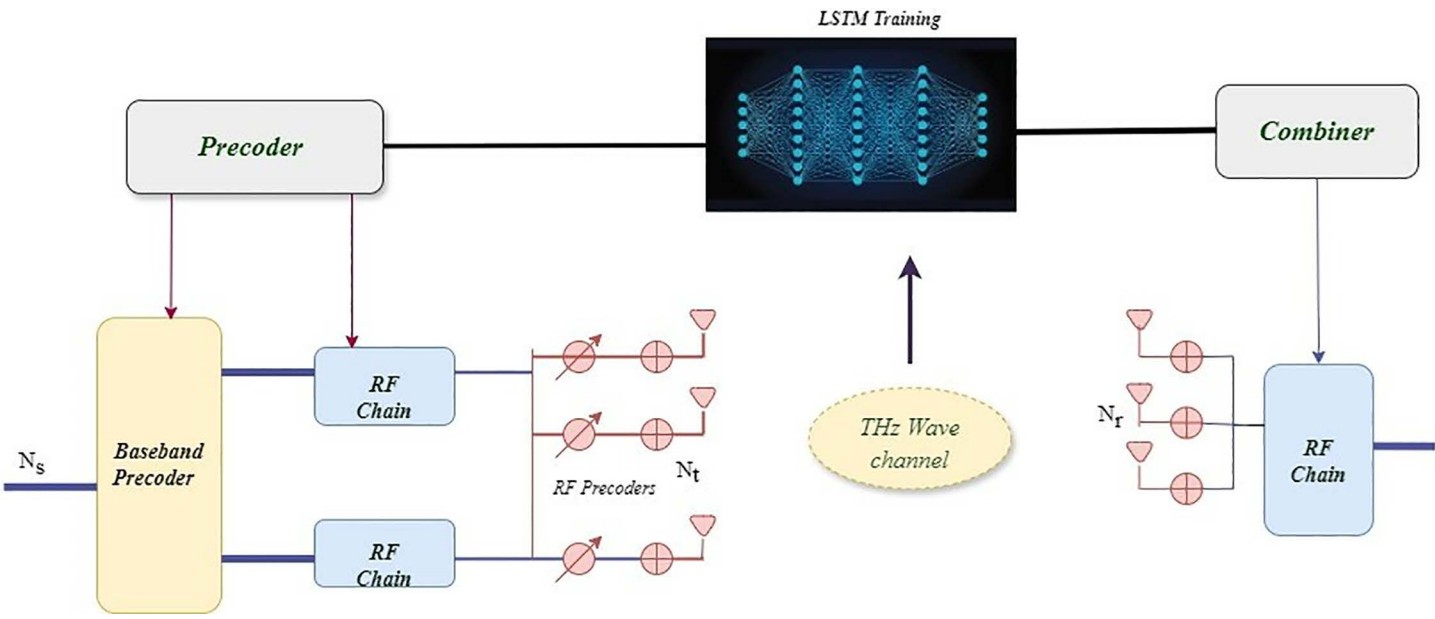

**Fig 2. RNN based hybrid precoding.**

The proposed RNN-based hybrid precoding method presents a significant advancement over traditional techniques such as Zero Forcing (ZF), Minimum Mean Square Error (MMSE), MRT (Maximal Ratio Transmission) and Alamouti schemes, particularly in the context of THz communication. Unlike ZF and MMSE, which are predominantly linear methods, the RNN-based approach leverages the adaptive and non-linear processing capabilities of recurrent neural networks, specifically Long Short-Term Memory (LSTM) networks, to optimize precoding weights dynamically. This dynamic adaptation is crucial for THz frequencies, which are characterized by rapid fluctuations in channel conditions due to high attenuation, limited scattering, and sensitivity to blockages. While ZF and MMSE work by solving linear equations to minimize interference and signal distortion, they are less effective in these high-frequency environments, as they fail to capture the temporal and spatial dependencies of the channel. The RNN-based method, on the other hand, continuously learns and updates its parameters based on past channel behavior, leading to more accurate channel estimation and better management of precoding weights, thus achieving superior spectral efficiency and BER performance [45].

Moreover, the comparison to Alamouti's space-time block coding highlights the differences between space diversity and the hybrid precoding approach. While Alamouti's method is a simple and effective scheme for achieving diversity in systems with two transmit antennas, it is limited in its ability to scale efficiently in massive MIMO configurations and does not fully exploit the benefits of hybrid analog-digital beamforming. The RNN-based hybrid precoding method not only scales with the increasing number of antennas in cell-free massive MIMO systems but also outperforms Alamouti and other conventional techniques by effectively combining both digital and analog precoding. This hybrid approach optimizes the balance between complexity and performance, reducing hardware constraints and power consumption while maintaining high throughput and low error rates in complex THz communication scenarios. The simulation results, showing the RNN-based method consistently achieving lower BER and higher spectral efficiency across a range of SNR values, further demonstrate its superiority over traditional methods like ZF, MMSE, and Alamouti, especially in next-generation wireless systems [46].

In this study, we implement and evaluate a RNN model for predicting hybrid precoding weights in dynamic wireless communication systems. Our approach leverages 70% of the dataset for training, followed by rigorous testing and

validation phases to ensure robust performance. The goal is to optimize spectral efficiency and BER in CFMM systems under THz communication conditions.

- **Training Phase:** The RNN is trained using 70% of the dataset composed of dynamic channel data and precoding information, with parameters such as real and imaginary channel gains, channel phases, and noise power. The ADAM (Adaptive Moment Estimation) optimizer is employed to minimize the MSE loss function, with a learning rate of 0.001 and a batch size of 128. The training process iteratively updates the RNN's weights to enhance its prediction accuracy.

- **Testing Phase:** After training, the RNN is tested on 15% of the separate dataset to evaluate its performance. This phase assesses the model's ability to generalize and accurately predict hybrid precoding weights under varying channel conditions, focusing on metrics like spectral efficiency and BER.

- **Validation Phase:** During training, 15% of the validation set is used to fine-tune the model and prevent overfitting. The RNN's performance on the validation set is monitored to adjust hyper parameters and ensure robust learning. The validation phase is crucial for optimizing the RNN's ability to adapt to real-time channel variations and maintain high spectral efficiency and low BER in CFMM systems under THz communication. This structured approach ensures that the RNN model is well-trained, accurately tested, and validated for reliable performance in dynamic wireless communication environments.

To tackle the intricate demands of CFMM systems in THz communication, a specialized RNN configuration is crafted. Key components include the ADAM optimizer, chosen for its adaptability to complex optimization landscapes, the TanH activation function for capturing nonlinear relationships in hybrid precoding, and the Mean Squared Error (MSE) loss function, prioritizing precise channel estimation crucial in THz communication. This tailored setup aims to enhance performance and reliability by efficiently handling large datasets, ensuring compatibility with real-world system requirements, and minimizing errors in predicted channel characteristics, thus enabling accurate precoding decisions in dynamic communication scenarios [47,48].

Adaptive coding and modulation play a pivotal role in enhancing the performance and resilience of MIMO systems, particularly in challenging communication environments like those encountered at higher frequencies such as the THz band. These techniques dynamically adjust the coding rate and modulation scheme based on real-time channel conditions, ensuring optimal data transmission while maintaining reliability. By adapting to fluctuating channel quality, including variations in SNR and interference levels, adaptive coding and modulation help to maximize spectral efficiency and minimize BER. This flexibility is crucial for MIMO systems operating in harsh conditions, where fixed modulation or coding schemes often fail to maintain reliable communication. The integration of these adaptive techniques encourages the design of more robust communication systems capable of withstanding the inherent challenges of high-frequency transmission, such as attenuation and fading, while ensuring efficient data throughput and low latency in next-generation wireless networks [49].

After Section 6, which integrates machine learning techniques to enhance CFMM system performance, Section 7 discusses the results and implications of our proposed approaches.

## 7 Result and discussion

The hybrid precoder simulation parameters for the study include several key elements. The simulation parameters were carefully chosen to reflect realistic conditions for THz communication and CFMM systems, ensuring a comprehensive evaluation of the proposed hybrid precoding scheme. In our study, we use large-scale antenna arrays to support the high demands of these systems. The number of transmit antennas $N_{TR}$ is set to 256, while the number of receive antennas $N_{RT}$ is 64. The simulation involves 5 paths $N_P$ and 10 clusters $N_C$. The operating frequency is 1 THz. For the angle of arrival (AoA) for clusters, represented as $\boldsymbol{a}_r(\varnothing_{k,il}^r, \theta_{k,il}^r)$, and the angle of departure (AoD) for paths, represented as $\boldsymbol{a}_r(\varnothing_{k,il}^t, \theta_{k,il}^t)$, values are randomly generated between 0 and $2\pi$. Channel gains for clusters and paths, denoted as $a_{ik,il}$, follow a

complex normal distribution with a mean of 0 and variance of 1, $CN(0,1)$. The channel matrix $\boldsymbol{H}_k$ is calculated based on the number of transmit and receive antennas, as well as the channel gains. The noise variance $\sigma^2$ is set at 10 dB. The simulation considers 10 users ($K$) and a transmit power ($P$) of 30 dB.

To ensure the reliability of the simulation results, an extensive dataset was used for the machine learning training. For each setting, metrics such as spectral efficiency and BER were averaged across all users and all evaluations. This methodology ensures statistically meaningful results and minimizes the impact of randomness or outlier behaviors. To validate the effectiveness of the proposed approach, its performance was benchmarked against a conventional precoding method without machine learning assistance. Results demonstrate that the RNN-based scheme consistently outperforms traditional methods in both sum rate and BER metrics, especially at higher user densities, confirming the model's adaptability to multi-user environments.

Fig 3, depicting spectral efficiency versus signal to noise ratio (SNR), aligns well with theoretical expectations and the hybrid precoder parameters discussed earlier in this paper. As SNR increases, the spectral efficiency improves across all methods, with the RNN-based approach consistently outperforming traditional methods such as MMSE, ZF, MRT, and Alamouti. Notably, the RNN-based method achieves the highest spectral efficiency across the entire SNR range from -20 dB to 30 dB, showcasing its superiority, particularly at higher SNR values where traditional methods converge towards similar efficiency levels. At 30 dB, the RNN-based method achieves 10 bps/Hz, while MMSE achieves 8.2 bps/Hz, ZF achieves 7.9 bps/Hz, MRT achieves 7.8 bps/Hz, and Alamouti has the least performance at 7.6 bps/Hz. This outcome validates the anticipated performance benefits of the RNN-based approach, especially when considering the hybrid precoding parameters designed to optimize spectral efficiency in a THz-wave CFMM system. Moreover, these results incorporate the contributions of an RNN-trained model, which is trained based on the parameters listed in Table 1. This model, integrated with the hybrid precoding parameters detailed in Algorithm 1, further enhances the system's performance. The integration of the RNN model with the hybrid precoder parameters exemplifies a robust approach to optimizing spectral efficiency,

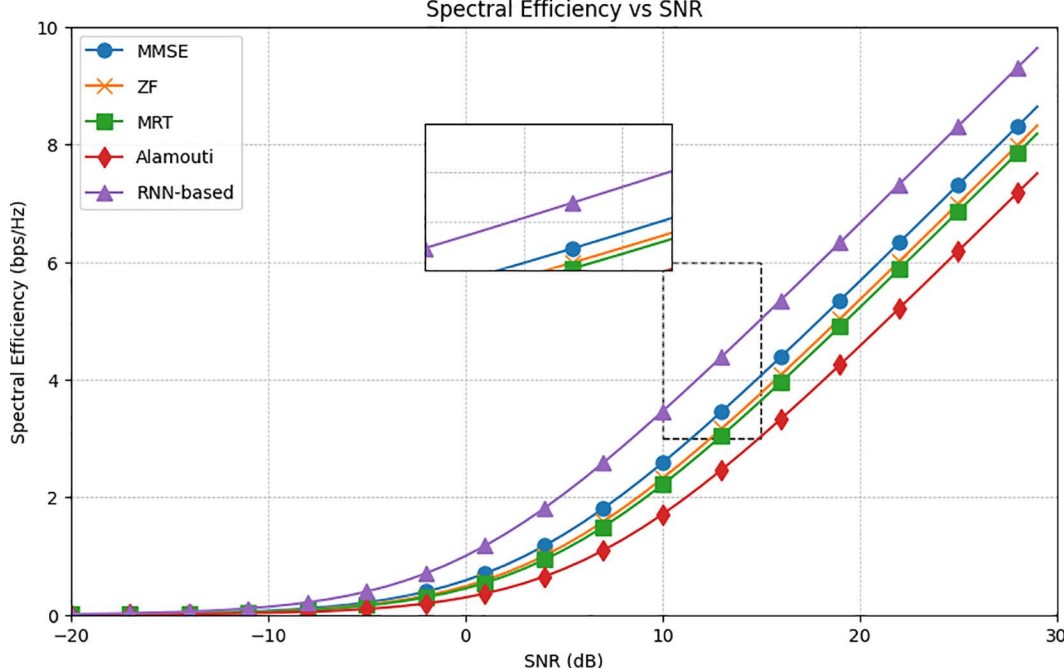

**Fig 3. The average spectral efficiency versus SNR.**

**Table 1. Simulation specifications for RNNs model training.**

| Parameter | Values |
|---|---|
| Input Layer neurons | 19 |
| Output Layer neurons | 6 |
| Hidden Layer 1neurons | 512 |
| Hidden Layer 2 neurons | 256 |
| Hidden Layer 3 neurons | 128 |
| Learning Rate | 0.001 |
| Optimizer | ADAM |
| Batch Size | 128 |
| Activation Function | Tanh |
| Loss Function | MSE |

demonstrating that the RNN-based method not only meets but exceeds theoretical predictions. This confirms its effectiveness in practical applications and highlights its potential for advancing THz wave communication systems.

Fig 4 illustrates the relationship between spectral efficiency and the number of transmitter antennas ($N\_TR$), with values extending up to 256 antennas. The RNN-based method consistently outperforms other techniques, achieving the highest spectral efficiency across all antenna configurations. Notably, as the number of transmitting antennas increases, the performance advantage of the RNN-based approach becomes even more evident, emphasizing its effectiveness in leveraging large antenna arrays for improved spectral efficiency. At 256 antennas, the RNN-based method reaches a spectral efficiency of 37 bps/Hz, followed by MMSE at 34 bps/Hz, ZF at 32 bps/Hz, MRT at 27 bps/Hz, and the Alamouti scheme at the lowest with 24 bps/Hz. Moreover, the integration of the RNN model with hybrid precoder parameters

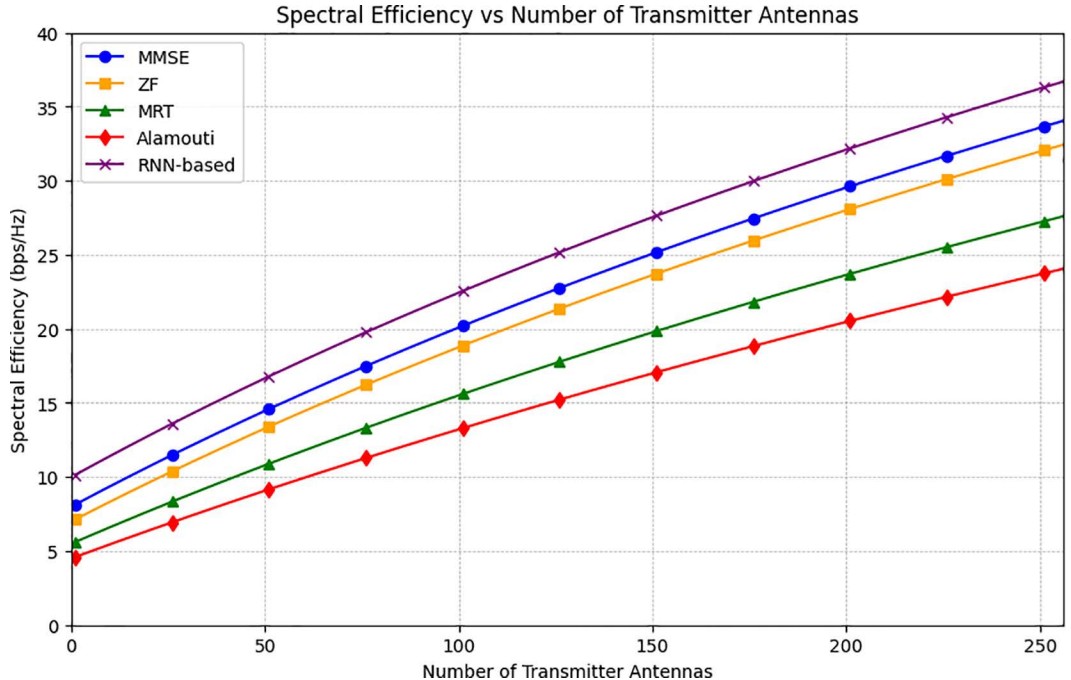

**Fig 4. The average spectral efficiency versus number of transmitter antennas ($N_{TR}$).**

confirms that the RNN-based method aligns with theoretical expectations. Specifically, equations (14) and (17), which depend on the number of transmitting antennas, are used to calculate spectral efficiency according to equation (8). This relationship underscores the RNN-based method's capacity to meet theoretical predictions while optimizing performance in THz-wave communication systems. The results demonstrate that as the number of transmitter antennas increases, so does the RNN-based method's ability to enhance spectral efficiency, making it a strong candidate for mMIMO systems with up to 256 antennas. Furthermore, the improved spectral efficiency is directly related to both the hybrid precoding parameters and the parameters involved in training the RNN model. The model exemplifies a powerful approach to optimizing spectral efficiency in large-scale antenna systems. However, the results would benefit from a clearer presentation to better highlight the relationship between these variables and the practical implications of the findings. The analysis leverages an extensive machine learning dataset to ensure statistical reliability, with results closely aligning with theoretical models and highlighting practical scalability for large mMIMO and THz systems.

Fig 5 illustrates the relationship between spectral efficiency and the number of users for various transmission schemes with up to 50 users. The RNN-based method consistently achieves the best performance across all user counts. As the number of users increases, the performance advantage of the RNN-based method becomes more pronounced, demonstrating its superior capability to efficiently serve multiple users. At 50 users, the RNN-based method achieves a spectral efficiency of 5.6 bps/Hz, while the MMSE method follows closely behind at 5 bps/Hz. The ZF and MRT schemes show similar performances, achieving 4.7 bps/Hz and 4.6 bps/Hz, respectively, with the Alamouti scheme having the lowest performance at 3.9 bps/Hz. These results validate the model's design and its ability to perform as expected, aligning with theoretical predictions. The increase in spectral efficiency with the number of users from 0 to 50 aligns with Equation 6, where the SINR depends on the number of users, directly affecting spectral efficiency. However, there is a specific point where the spectral efficiency does not significantly increase with the number of users. This plateau effect arises from interference and limited channel resources, which become more prominent as user density increases, thereby capping

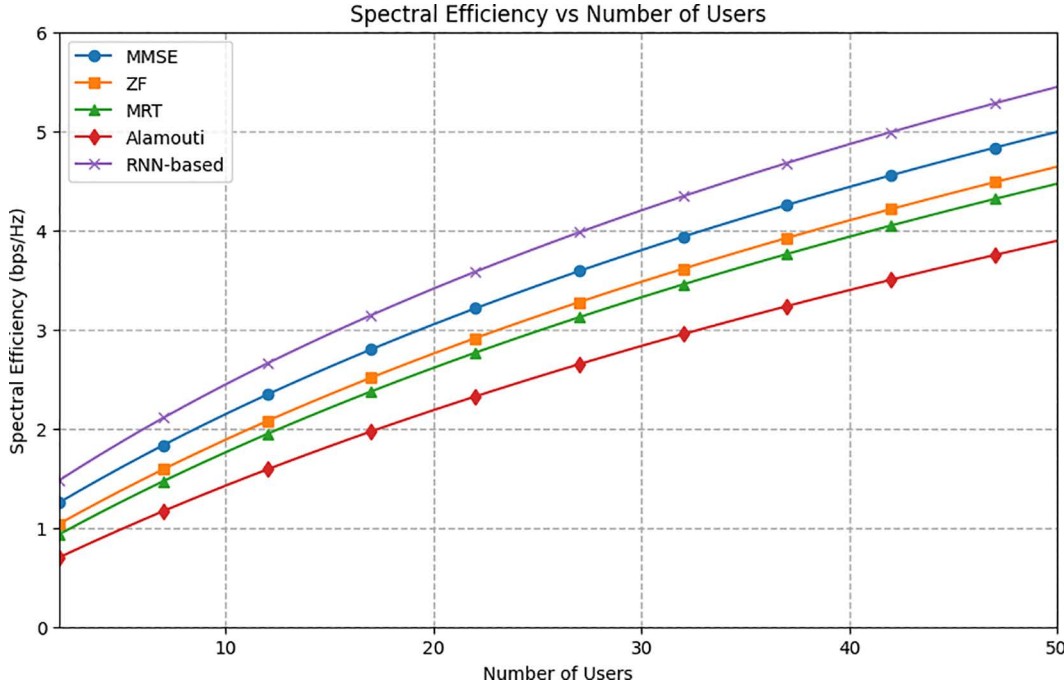

**Fig 5. The average spectral efficiency versus number of users.**

the achievable spectral efficiency typical of advanced transmission techniques. It is important to clarify that Fig 5 reports the *total spectral efficiency (SE)* across all users, computed as the sum of individual user SEs, $\sum_{k=1}^{K} SE_k$. While it is true that increasing the number of users generally leads to reduced SINR and thus lower individual SEs, the total SE may still increase because it accumulates contributions from a larger user base. This trend reflects the enhanced multi-user capability of CFMM systems, where the aggregate system throughput benefits from user multiplexing, despite per-user trade-offs.

Fig 6 illustrates the relationship between spectral efficiency and the number of RF chains for various transmission schemes in a massive MIMO (mMIMO) system, with RF chains ranging from 4 to 8. The RNN-based technique consistently demonstrates the highest spectral efficiency across all RF chain configurations. At the upper end, with 8 RF chains, the RNN-based method achieves a spectral efficiency of 35.5 bps/Hz, outperforming other methods such as MMSE (33 bps/Hz), ZF (31 bps/Hz), MRT (30 bps/Hz), and Alamouti (25 bps/Hz). These results highlight the superior performance of the RNN-based method, particularly as the number of RF chains increases, showcasing its effectiveness in optimizing performance for mMIMO systems with up to 8 RF chains. The RNN-based approach benefits significantly from an increasing number of RF chains, as it allows the system to process more data streams through improved analog and digital precoding weights. This is explained in Section 3, where the fully connected nature of the system and the inclusion of additional RF chains enable enhanced data processing capabilities.

The precoding weights, as defined in Equations 14 and 18, play a crucial role in this process by optimizing how signals are transmitted, leading to higher spectral efficiency. The integration of the RNN model with hybrid precoding, as depicted in Fig 2, further refines this process, dynamically adjusting the precoding to optimize system performance. However, it is essential to consider the tradeoff between the number of RF chains and the associated costs and complexity. While increasing the number of RF chains can significantly improve spectral efficiency, this comes with a higher hardware cost and greater computational demands due to the need for more complex processing. Additionally, the improvements in

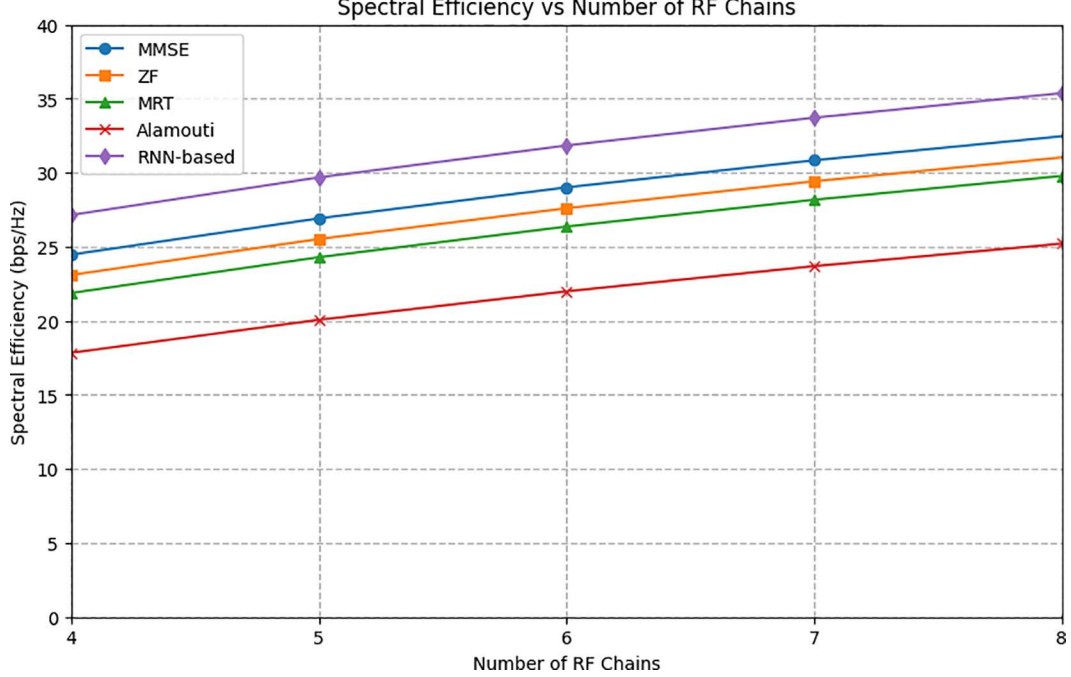

**Fig 6. The average spectral efficiency versus number of RF chains.**

spectral efficiency do not continue indefinitely as the number of RF chains increases. Beyond a certain threshold, the system may experience diminishing returns, where adding more RF chains yields minimal gains due to factors such as interference and the limitations of spatial multiplexing gains.

While SU-MMIMO can achieve high spectral efficiency for a single user by dedicating all resources, it is inherently limited in multiuser scenarios. This study focuses on CFMM, which is specifically designed to serve multiple users simultaneously through distributed APs and hybrid precoding techniques. The proposed RNN-based hybrid precoding method dynamically optimizes both analog and digital precoders, effectively mitigating inter-user interference and enhancing spectral efficiency across multiple users. As shown in Figs 3–6, CFMM consistently outperforms traditional methods in multiuser setups, making it more practical for dense and dynamic environments where SU-MMIMO would struggle to equitably allocate resources and maintain performance. Thus, while SU-MMIMO may excel for isolated single-user cases, CFMM is better suited for scalable, next-generation wireless networks such as those operating in the THz band [50].

Fig 7 illustrates the BER performance as a function of SNR for 16-QAM under various detection schemes in a system with 256 transmitting antennas. The RNN-based approach demonstrates the best performance, achieving a BER of $10^{-4}$ at 20.5 dB. The MMSE method closely follows, attaining the same BER at 21 dB, while ZF requires 21.5 dB to achieve the same performance. On the other hand, MRT and Alamouti techniques exhibit the least efficiency, reaching $10^{-4}$ at 22 dB and 23 dB, respectively. These results validate the high feasibility of the RNN algorithm in suppressing detection errors within CFMM systems. The superior performance of the RNN-based system stems from the integration of an RNN-trained model with appropriately selected hybrid precoding parameters. This combination significantly contributes to BER reduction by dynamically adjusting precoding weights, as outlined in Algorithm 1, optimizing spectral efficiency, and

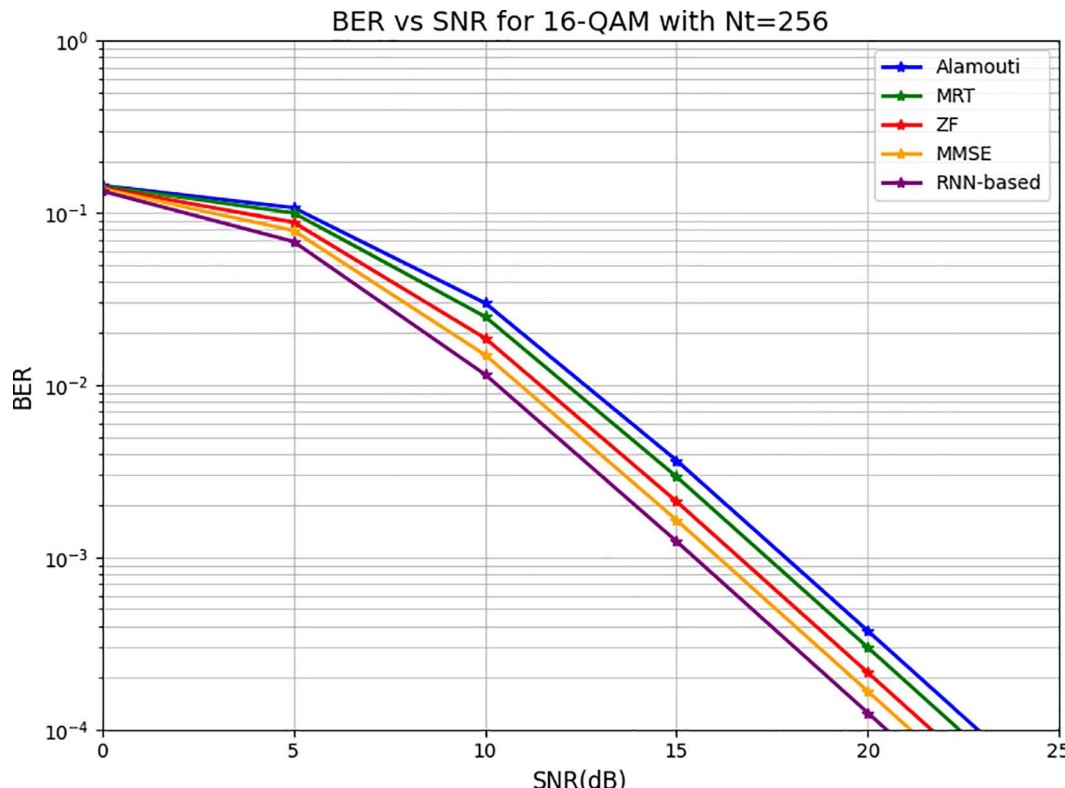

**Fig 7. The BER versus SNR for 16-QAM with 256 transmitting antennas.**

minimizing detection errors, as demonstrated in Equation 6. Moreover, the figure highlights that the RNN-based approach consistently outperforms alternative methods, especially in the SNR range of 0–15 dB. The observed linear decay in the BER plot as SNR increases is attributed to the progressive enhancement of signal quality relative to noise power. As SNR grows, the error probability diminishes because higher SNR levels lead to more distinct constellation points in 16-QAM, reducing symbol misclassification. Additionally, the RNN's ability to dynamically adapt the precoding weights further ensures efficient interference mitigation, allowing BER to decrease steadily with rising SNR.

Fig 8 showcases the BER performance as a function of SNR for 64-QAM modulation under various detection schemes in a system with 256 transmitting antennas. The RNN-based approach achieves a BER of $10^{-4}$ at 24 dB, marginally out-performing MMSE, which reaches the same BER at 25 dB. Other detection schemes, including ZF, MRT, and Alamouti, follow sequentially, requiring progressively higher SNRs to achieve equivalent BER performance. A critical observation is that the BER curves for 64-QAM decline at a noticeably slower rate compared to those for 16-QAM, emphasizing the greater sensitivity of 64-QAM to noise and signal distortions. This heightened sensitivity underscores the trade-off inherent in higher-order modulation schemes: while 64-QAM enables increased data rates due to its ability to transmit more bits per symbol, it is substantially less robust against noise. This trade-off poses a significant challenge in practical scenarios, especially in environments with fluctuating or suboptimal SNR conditions. The BER curves shown are the average BER across all UEs, computed under varying transmit power conditions while maintaining the same scenario configuration. the integration of the RNN model with the hybrid precoder demonstrates substantial performance gains, enabling dynamic adjustment of precoding weights and enhanced BER reduction. Additionally, while the BER decay signifies rapid improvements in detection performance with increasing SNR, it also reflects the fragility of 64-QAM at lower SNR levels, where

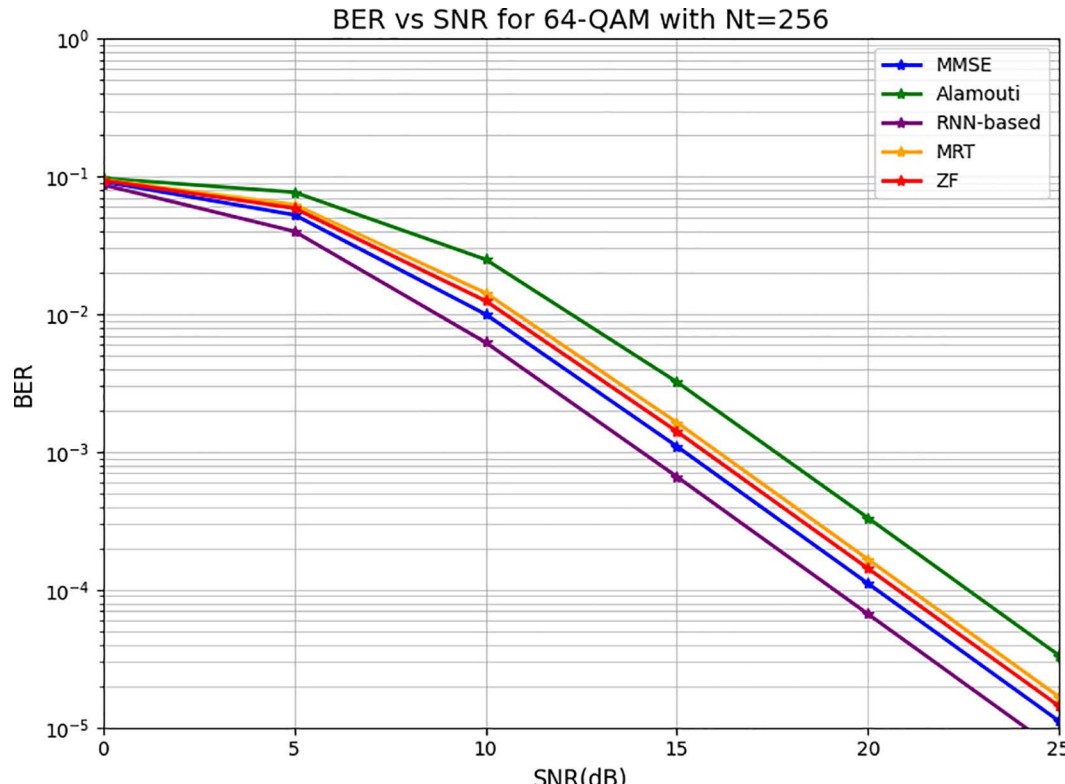

**Fig 8. The BER versus SNR for 64-QAM with 256 transmitting antennas.**

error rates are significantly higher. This analysis indicates that while the RNN-based approach holds promise for advancing CFMM systems, further exploration of its adaptability and resilience in diverse operating conditions is necessary. The complexity of 64-QAM, coupled with the marginal gains over traditional methods like MMSE, calls for a deeper evaluation of the computational trade-offs and real-world applicability of the RNN-based model in large-scale deployments.

Fig 9 illustrates the BER performance as a function of SNR for 64-QAM modulation in a system with 64 transmitting antennas across various detection schemes. The RNN-based approach achieves a BER of $10^{-4}$ at 25 dB, outperforming other methods such as MMSE, ZF, MRT, and Alamouti, which require progressively higher SNRs to reach the same BER level. This result raises critical questions regarding the effectiveness of the 64-antenna configuration in ensuring robust communication performance. The reduced number of antennas limits the diversity gain and spatial resolution achievable by the system, which directly impacts the BER performance of the employed precoding techniques. Comparatively, in systems with higher antenna counts, such as the previously analyzed 256-antenna configuration, all detection methods demonstrate significantly lower BERs for the same modulation scheme. This improvement underscores the advantages of increased antenna diversity, which enhances signal processing capabilities and resilience against noise and interference. The integration of the RNN model with the hybrid precoder remains pivotal in achieving these BER reductions. By dynamically adjusting the precoding weights and optimizing the system parameters, as outlined in Algorithm 1, the RNN-based approach continues to validate its effectiveness in addressing the challenges posed by complex modulation schemes like 64-QAM, even in lower-antenna configurations. However, the performance gap between 64 and 256 antennas emphasizes the need for further refinement and optimization of the system to fully exploit its potential in practical scenarios.

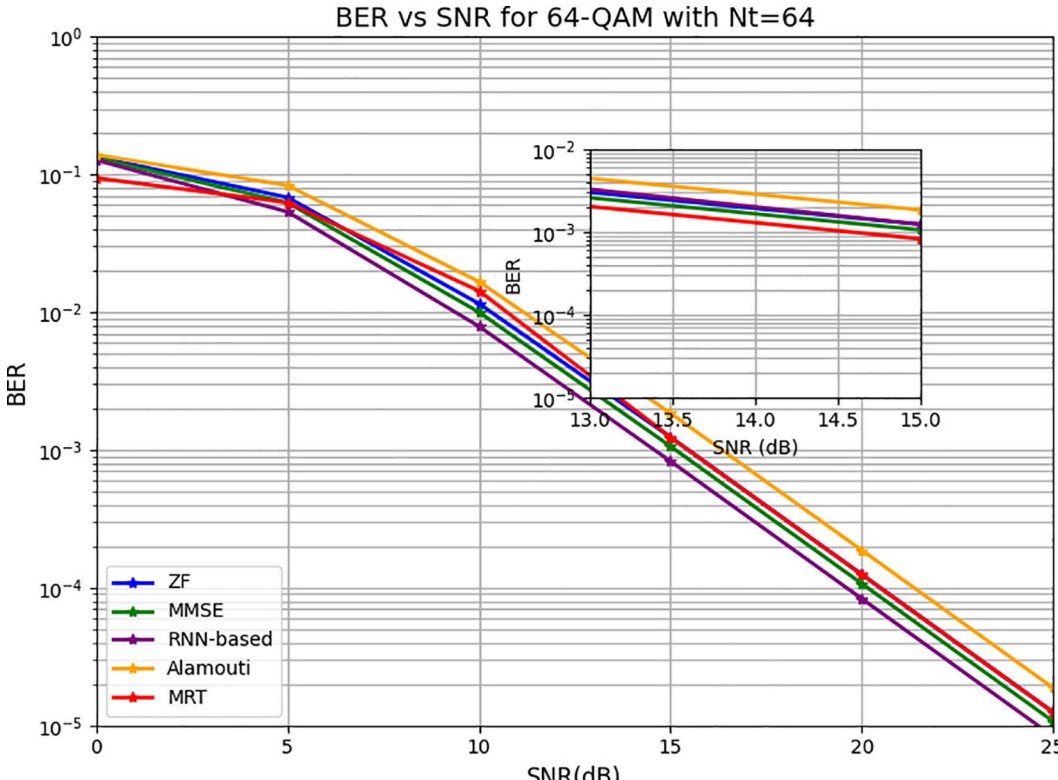

**Fig 9. The BER versus SNR for 64-QAM for 64 transmitting antennas.**

Turbo codes are well-known for their ability to improve BER through iterative error correction, making them highly effective in mitigating noise and interference in wireless communication. However, in this study, the focus is on evaluating the performance of the proposed RNN-based hybrid precoding method, particularly in optimizing precoding weights to adapt to the unique challenges of THz communication. While turbo codes could further enhance BER by improving error resilience, their integration was beyond the scope of this work, as it would require additional computational complexity and changes to the system architecture. Future research will explore the combination of advanced channel coding schemes like turbo codes with the RNN-based precoding approach to achieve even greater BER improvements in THz communication systems [51].

The proposed RNN-based precoder adapts effectively to the dynamic channel conditions, enabling a significant improvement in average spectral efficiency across different transmit power levels. The recurrent nature of the model allows it to track slow variations in THz channels, which are typically dominated by line-of-sight and a few scatterers, leading to more accurate beamforming and reduced inter-user interference.

The scalability and efficiency of the proposed method make it a strong candidate for practical deployment in future THz-based indoor access points and small-cell networks. The RNN structure can be efficiently implemented with modern edge computing resources, minimizing the latency introduced by precoding updates in real-time communications

The proposed work, while innovative and promising, has some limitations, which need to be addressed in future research.

- **Computational Complexity:** The method introduces significant computational demands, especially with large-scale antenna arrays and high-dimensional data in THz communication. Real-time processing may be difficult, requiring optimization strategies to make the approach more practical for deployment.

- **Channel Estimation Accuracy:** The effectiveness of the RNN-based hybrid precoding heavily relies on accurate channel state information (CSI). Inaccurate CSI can degrade system performance, leading to suboptimal spectral efficiency and increased BER. Enhancing channel estimation techniques is crucial to mitigating these issues.

- **Energy Efficiency:** The power consumption of implementing RNNs and hybrid precoding at THz frequencies has not been thoroughly explored. Ensuring energy-efficient operation is crucial for sustainable deployment in large-scale wireless networks.

## 8 Conclusions

In CFMM systems operating within the THz band, the adoption of a hybrid precoding strategy based on RNNs has emerged as highly effective under challenging conditions. Simulation results demonstrate clear advantages over traditional precoding techniques. specifically, the RNN-based approach achieves superior spectral efficiency and reduced BER. Under varying SNR, the RNN configuration achieves 10 bps/Hz compared to 8.2 bps/Hz with MMSE, showcasing its ability to dynamically adjust analog and digital precoders for optimal performance. Moreover, in scenarios with multiple users, the RNN approach achieves 5.6 bps/Hz, surpassing the MMSE's 5 bps/Hz, indicating robust handling of dense user environments and high antenna counts crucial for future networks. For BER performance with 16-QAM modulation, the RNN-based method achieves a BER of $10^{-6}$ at 10 dB SNR, ensuring reliable communication amidst the challenges of THz frequencies such as high attenuation and limited scattering. Moving forward, future research should explore scalability in terms of increasing antenna and user numbers, adaptive strategies for diverse wireless environments, and the transition from simulation to practical implementation to validate these promising results in real-world settings and also Future work will focus on extending the RNN architecture to support user-centric cell-free mMIMO scenarios with dynamic AP selection, aiming to further enhance fairness among UEs and address quality-of-service (QoS) differentiation. Additionally, hardware implementation feasibility studies for RNN-based precoding at THz bands will be conducted. Overall, the

integration of RNN-based hybrid precoding marks a significant step toward meeting the stringent demands of next-generation wireless networks in THz bands.

## Supporting information

**S1 Data. Dataset file** . Excel file containing the dataset used for model training in the study.
(XLSX)

**S2 Code. Python code** . Script used for generating the results presented in the manuscript.
(DOCX)

**S3 Text. README file** . Provides instructions and information on the purpose and usage of the accompanying code.
(TXT)

## Author contributions

**Conceptualization:** Tadele A. Abose, Binyam G. Assefa, Yitbarek A. Mekonen, Naol W. Gudeta.

**Methodology:** Tadele A. Abose, Binyam G. Assefa, Yitbarek A. Mekonen, Naol W. Gudeta.

**Software:** Tadele A. Abose, Binyam G. Assefa, Yitbarek A. Mekonen, Naol W. Gudeta.

**Supervision:** Tadele A. Abose.

**Validation:** Tadele A. Abose, Binyam G. Assefa, Yitbarek A. Mekonen, Naol W. Gudeta.

**Visualization:** Tadele A. Abose, Binyam G. Assefa, Yitbarek A. Mekonen, Naol W. Gudeta.

**Writing – original draft:** Tadele A. Abose, Binyam G. Assefa, Yitbarek A. Mekonen, Naol W. Gudeta.

**Writing – review & editing:** Tadele A. Abose, Binyam G. Assefa, Yitbarek A. Mekonen, Naol W. Gudeta.

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
