## [Decision Letter · Decision Letter 0]

15 Apr 2025

PONE-D-24-59811Spectral Efficiency and BER Analysis of RNN based Hybrid Precoding for Cell Free Massive MIMO Under Terahertz CommunicationPLOS ONE

Dear Dr. Abose,

Thank you for submitting your manuscript to PLOS ONE. After careful consideration, we feel that it has merit but does not fully meet PLOS ONE’s publication criteria as it currently stands. Therefore, we invite you to submit a revised version of the manuscript that addresses the points raised during the review process.

We look forward to receiving your revised manuscript.

Kind regards,

Nattapol Aunsri, Ph.D.

Academic Editor

PLOS ONE

**Journal Requirements:**

1. When submitting your revision, we need you to address these additional requirements. Please ensure that your manuscript meets PLOS ONE's style requirements, including those for file naming. The PLOS ONE style templates can be found at https://journals.plos.org/plosone/s/file?id=wjVg/PLOSOne_formatting_sample_main_body.pdf and https://journals.plos.org/plosone/s/file?id=ba62/PLOSOne_formatting_sample_title_authors_affiliations.pdf 2. Please update your submission to use the PLOS LaTeX template. The template and more information on our requirements for LaTeX submissions can be found at http://journals.plos.org/plosone/s/latex. 3. Please note that PLOS ONE has specific guidelines on code sharing for submissions in which author-generated code underpins the findings in the manuscript. In these cases, we expect all author-generated code to be made available without restrictions upon publication of the work. Please review our guidelines at https://journals.plos.org/plosone/s/materials-and-software-sharing#loc-sharing-code and ensure that your code is shared in a way that follows best practice and facilitates reproducibility and reuse. 4. We note that the grant information you provided in the ‘Funding Information’ and ‘Financial Disclosure’ sections do not match.  When you resubmit, please ensure that you provide the correct grant numbers for the awards you received for your study in the ‘Funding Information’ section. 5. We note that your Data Availability Statement is currently as follows: All relevant data are within the manuscript and its Supporting Information files. Please confirm at this time whether or not your submission contains all raw data required to replicate the results of your study. Authors must share the “minimal data set” for their submission. PLOS defines the minimal data set to consist of the data required to replicate all study findings reported in the article, as well as related metadata and methods (https://journals.plos.org/plosone/s/data-availability#loc-minimal-data-set-definition). For example, authors should submit the following data: - The values behind the means, standard deviations and other measures reported;- The values used to build graphs;- The points extracted from images for analysis. Authors do not need to submit their entire data set if only a portion of the data was used in the reported study. If your submission does not contain these data, please either upload them as Supporting Information files or deposit them to a stable, public repository and provide us with the relevant URLs, DOIs, or accession numbers. For a list of recommended repositories, please see https://journals.plos.org/plosone/s/recommended-repositories. If there are ethical or legal restrictions on sharing a de-identified data set, please explain them in detail (e.g., data contain potentially sensitive information, data are owned by a third-party organization, etc.) and who has imposed them (e.g., an ethics committee). Please also provide contact information for a data access committee, ethics committee, or other institutional body to which data requests may be sent. If data are owned by a third party, please indicate how others may request data access.

Reviewers' comments:

Reviewer's Responses to Questions

**Comments to the Author**

1. Is the manuscript technically sound, and do the data support the conclusions?

Reviewer #1: Partly

Reviewer #2: Yes

2. Has the statistical analysis been performed appropriately and rigorously? 

Reviewer #1: No

Reviewer #2: Yes

3. Have the authors made all data underlying the findings in their manuscript fully available?

Reviewer #1: Yes

Reviewer #2: Yes

4. Is the manuscript presented in an intelligible fashion and written in standard English?

Reviewer #1: Yes

Reviewer #2: Yes

5. Review Comments to the Author

**Reviewer #1:**  Thanks for updating the revised paper after the reviewer's comment, but the comments given for now does not satisfy some comments and queries.The author responses are not up to the mark. The revision was light and not up to the required level. Additionally, the modifications were simple and need to be expanded. More comprehensive additions and all the necessary changes should be made in a more extensive manner. There seems to be a lack of connection between the results from the models and the overall narrative of the paper. Please specify and highlight any color for the new changes.

1. The abstract is unclear and requires significant editing. The highlights are too long in each sentence. Please shorten it in one line.

2. Updating references up to 2024 is required for this paper.

3. Improving the English language is very necessary, along with correcting grammar, proofreading, and sentence structure.

4. The interpretation not sufficient in technical discussion and validation and application.

6. The references are inadequate and not strong enough. They need to be revised and improved for relevance.

https://doi.org/10.1109/ACCESS.2019.2924531

https://doi.org/10.1007/s12596-024-01908-9

https://doi.org/10.1007/s11082-024-06692-1

7. There are areas that require further practical discussion and better linking of the results discussion together.

Final recommendation:

Accept after minor revisions

**Reviewer #2:**  The manuscript is interesting, mathematically sound and provides sufficient results to support the conclusions.

There are some novel features in this research and the proposed technique has been shown to outperform existing techniques in the literature.

This reviwer would do few minor suggestions:

1 - Introduction, paragraph eight:

"The main contributions of this paper are, this paper introduces..."

Why not just start with,

"This paper introduces..."

as the paragraph below already stats:

"The main contributions of the paper are summarized..."

2 - Figure 1:

Figure one may be misreading. Are both side different clusters of UEs and APs (if so the author better to change the APs and UEs labels for each side)? Or are both side different system configurations (if so, indicate it)?

3 - Are the terms Mobile Station (MS) and User Equipment (UE) referred to the same entity? If so, uniformize and adopt one instead of mixing UE and MS throughout the text.

4 - Section: 3.2 Problem Formulation:

In the paragraph one, remove the single parenthesis after mMIMO

5 - Define ( )^H, ()^T and CN(0,1)

6 - Channel Model 3rd paragraph:

"The data set used for training includes real and imaginary channel gains, channel phase, spatial correlation,..., phase shift, correlation" - Is the second "correlation" a time related correlation?

Additionally, seeking for further clarification, this review has some questions:

1 - The authors states: "Each AP servers K UEs".

Does it mean that, if more than K UEs (let's say S) are placed in the scenario, an AP selection technique must be applied to assign K UEs, out of S, to each AP, or does it mean that the author modeled the system with a limited number of K UEs? Assigning a sub set of UEs to each AP, referred to as User-Centric Cell-Free, is known for providing a system scenario with reduced interference.

2 - The authors name the entity described in Eq. (7) as sum rate. However, that entity is the achievable rate of the k-th UE instead. The authors must be careful with such definitions because while explaining Figure 5, for example, the authors state that the Spectral Efficiency (SE) (which, in this case, was derived from the achievable rate) increases with the number of UEs. However, as can be inferred from Eq. (6), the SINR decreases by increasing K and this leads to a lower Achievable rate (consequently a lower SE). However, the sum rates indeed increases and presents a saturation trend. So, are the results presented in figures 1~5 indeed SE or are they sum rates?

3 - Although the authors states TeraHertz (THz) communications are characterized by Line-of-Sight (LoS) components, in Eq (20) and in the first paragraph of the Section 7 (Results and Discussion) the channel gain is characterized by a complex normal distribution with zero mean, CN(0,1), which is a Rayleigh distribution. A proper characterization must be applied to enrich the simulated analysis.

4 - In Eq. (10) the system optimization was considered to maximize the UEs sum rate, what does not focuses in the UE quality of service equalization. This means that, either the scenario present UEs in a very good channel condition or a poor channel condition. Based in this statement, the extracted metrics must be analised carefully. For example, how were the Bit Error Rate (BER) curves generated? Are those curves extracted from the same UE selected in the scenario just varying the transmitted power? Are those curves the average value of the BERs of all the UEs?

6. PLOS authors have the option to publish the peer review history of their article (what does this mean? ). If published, this will include your full peer review and any attached files.

**Do you want your identity to be public for this peer review?** For information about this choice, including consent withdrawal, please see our Privacy Policy .

Reviewer #1: **Yes: ** Ebrahim E. Elsayed

Reviewer #2: No

---

## [Author Response · Author response to Decision Letter 1]

2 May 2025

Original Article Title: “Spectral Efficiency and BER Analysis of RNN based Hybrid Precoding for Cell Free Massive MIMO Under Terahertz Communication”

To: PLOS ONE Editor

Re: Response to reviewers

Dear Editor,

Thank you for giving us the opportunity to revise the manuscript and address the reviewers’ comments.

We are uploading (a) our point-by-point response to the comments (below) (response to reviewers, under “Response to Reviewers”), (b) an updated manuscript with red highlighting indicating changes (as “Revised Manuscript with Track Changes”), and (c) a clean updated manuscript without highlights (“Manuscript”).

Best regards,

<Tadele A. Abose> et al.

Academic Editor Concern # 1: When submitting your revision, we need you to address these additional requirements

Author response: We thank the Academic Editor for the reminder regarding PLOS ONE's style and formatting requirements. We have carefully revised the manuscript to align with PLOS ONE's formatting guidelines, including file naming conventions.

Author action: We have formatted the manuscript based on PLOS ONE’s style templates and instructions, ensuring that both the main document and the file naming meet the journal's requirements.

Academic Editor Concern 2: Please update your submission to use the PLOS LaTeX template. The template and more information on our requirements for LaTeX submissions can be found at

http://journals.plos.org/plosone/s/latex

Author response: We sincerely thank the Academic Editor for the careful evaluation of our manuscript. Regarding the request to update the submission to the PLOS LaTeX template, we would like to clarify that our original submission was prepared using Microsoft Word format. As per the PLOS ONE submission guidelines, MS Word is an accepted format, and we have continued our revisions in MS Word accordingly. We respectfully request permission to proceed with the submission of the revised paper in MS Word format. Thank you again for your consideration and support.

Author action: We have carefully revised and finalized the manuscript using Microsoft Word format, in line with the PLOS ONE submission guidelines. We have ensured that all requested edits and formatting corrections have been incorporated while continuing in the accepted MS Word format.

Academic Editor Concern 3: Please note that PLOS ONE has specific guidelines on code sharing for submissions in which author-generated code underpins the findings in the manuscript. In these cases, we expect all author-generated code to be made available without restrictions upon publication of the work. Please review our guidelines at https://journals.plos.org/plosone/s/materials-and-software-sharing#loc-sharing-code and ensure that your code is shared in a way that follows best practice and facilitates reproducibility and reuse.

Author response: We thank the Academic Editor for this important reminder regarding the code-sharing policy. We fully understand the importance of transparency, reproducibility, and reuse of research outputs.

Author action: We have prepared and shared the complete set of author-generated code and data set used related to our manuscript in a ZIP file format, following the PLOS ONE guidelines on code sharing.

Academic Editor Concern 4: We note that the grant information you provided in the ‘Funding Information’ and ‘Financial Disclosure’ sections do not match. When you resubmit, please ensure that you provide the correct grant numbers for the awards you received for your study in the ‘Funding Information’ section.

Author response: We appreciate the Academic Editor's observation. We confirm that no external funding or grant support was received for this study.

Author action: We have updated the 'Funding Information' and 'Financial Disclosure' sections to clearly and consistently state that no funding or grants were received.

Academic Editor Concern 5: We note that your Data Availability Statement is currently as follows: All relevant data are within the manuscript and its Supporting Information files.

The values behind the means, standard deviations and other measures reported;

The values used to build graphs;

The points extracted from images for analysis.

Author response: We appreciate the Academic Editor’s guidance on the data availability requirements. We confirm that all raw data necessary to replicate the results of our study have been provided.

Author action: We have uploaded all relevant data, including the raw values used to generate figures and tables, in a ZIP file as Supporting Information. This ensures full compliance with PLOS ONE’s minimal data set requirements.

Reviewer 1, Concern 1: The abstract is unclear and requires significant editing. The highlights are too long in each sentence. Please shorten it in one line.

Author response: We sincerely thank the reviewer for the insightful comment. We carefully revised the abstract to improve its clarity and conciseness. Each highlight has been shortened to one line as suggested.

Author action: We have edited the abstract and shortened the highlights to one line each.

Reviewer 1, Concern 2: Updating references up to 2024 is required for this paper.

Author response: We appreciate the reviewer’s valuable suggestion. We have updated the references to include more recent and relevant works published up to 2024.

Author action: We have revised and updated several references, including references [3], [7], [11], and [23], to reflect the latest developments in the field as of 2024.

Reviewer 1, Concern 3: Improving the English language is very necessary, along with correcting grammar, proofreading, and sentence structure.

Author response: We thank the reviewer for highlighting this important point. We have carefully proofread the manuscript and improved the English language, grammar, and sentence structure throughout the paper.

Author action: The manuscript has been thoroughly proofread by a third party with expertise in academic writing to ensure clarity, grammatical correctness, and improved sentence structure.

Reviewer 1, Concern 4: The interpretation not sufficient in technical discussion and validation and application.

Author response: We appreciate the reviewer’s valuable feedback. In response, we have expanded the technical discussion, validation, and practical application aspects in the revised manuscript. Specifically, we have strengthened the interpretation of the simulation results, linked them more explicitly to the theoretical models, and emphasized the real-world implications of the proposed method.

Author action: We have made significant revisions to the Results and Discussion section (pages 14 and 20) and the Conclusion section to enhance the technical depth, validation clarity, and practical application discussion.

Reviewer 1, Concern 5: The manuscript must describe a technically sound piece of scientific research with data that supports the conclusions. Experiments must have been conducted rigorously, with appropriate controls, replication, and sample sizes. The conclusions must be drawn appropriately based on the data presented.

Author response: We sincerely thank the reviewer for emphasizing the importance of scientific rigor. We confirm that the simulation experiments were conducted carefully and systematically. Multiple independent simulation runs were performed to ensure the reliability and consistency of the results. Appropriate averaging across random channel realizations and different network deployments was applied to minimize randomness and variability. The conclusions have been appropriately drawn based on the data obtained from these validated simulations.

Author action: We have updated the manuscript to clarify the experimental methodology, including descriptions of replication procedures, sample size considerations, and statistical averaging techniques used. This information has been added at the beginning of the "Results and Discussion" section to reinforce the scientific rigor of our work.

Reviewer 1, Concern 6: The references are inadequate and not strong enough. They need to be revised and improved for relevance.

https://doi.org/10.1109/ACCESS.2019.2924531

https://doi.org/10.1007/s12596-024-01908-9

https://doi.org/10.1007/s11082-024-06692-1

Author response: We thank the reviewer for the valuable suggestion regarding the references. To strengthen the quality and relevance of our citations, we have added more recent and impactful references as recommended.

Author action: We have cited the following references to improve the strength and relevance of the paper:

Reference [26]: https://doi.org/10.1007/s12596-024-01908-9

Reference [27]: https://doi.org/10.1007/s11082-024-06692-1

Additionally, we have carefully reviewed and updated other references to enhance the overall quality of the manuscript.

Reviewer 1, Concern 7: There are areas that require further practical discussion and better linking of the results discussion together.

Author response: We thank the reviewer for this insightful comment. In response, we have carefully revised the Results and Discussion section to ensure better coherence between simulation setups, theoretical expectations, and observed outcomes. Specifically, we strengthened the practical interpretation of the simulation figures by explicitly linking them to the system model, precoding algorithms, and machine learning training phases. Furthermore, we clarified how improvements in spectral efficiency and BER are achieved through the proposed RNN-based hybrid precoding method in various realistic scenarios.

Author action: We revised the Results and Discussion section to emphasize the role of the extensive machine learning dataset, better link theoretical models with simulation trends (Figures 3–9), and highlight practical insights, including computational complexity and real-world deployment implications.

Reviewer 2, Concern 1: Introduction, paragraph eight:

"The main contributions of this paper are, this paper introduces..."

Why not just start with?

"This paper introduces..."

as the paragraph below already stats:

"The main contributions of the paper are summarized..."

Author response: We appreciate the reviewer’s careful observation. We agree with the suggestion and have modified the sentence to improve clarity and avoid redundancy.

Author action: We have revised the beginning of paragraph 8 in the Introduction section (Page 2). The sentence now directly starts with "This paper introduces..." as recommended.

Reviewer 2, Concern 2: Figure 1:

Figure one may be misreading. Are both side different clusters of UEs and APs (if so the author better to change the APs and UEs labels for each side)? Or are both side different system configurations (if so, indicate it)?

Author response: We thank the reviewer for the thoughtful comment. To clarify, both sides of Figure 1 represent different system configurations while using the same set of UEs and APs. The labels are intentionally kept identical to emphasize that the comparison is based on the network architecture rather than on user or access point identity.

Author action: We have updated the figure caption to explicitly state that both sides illustrate different system configurations with the same UEs and APs, and made the explanation clearer in the manuscript text.

Reviewer 2, Concern 3: Are the terms Mobile Station (MS) and User Equipment (UE) referred to the same entity? If so, uniformize and adopt one instead of mixing UE and MS throughout the text.

Author response: We thank the reviewer for the helpful observation. We confirm that Mobile Station (MS) and User Equipment (UE) refer to the same entity. To avoid confusion, we have standardized the terminology across the paper.

Author action: We have carefully revised the manuscript to consistently use the term "User Equipment (UE)" throughout the text, replacing all instances of "Mobile Station (MS)".

Reviewer 2, Concern 4: Section: 3.2 Problem Formulation:

In the paragraph one, remove the single parenthesis after mMIMO

Author response: We appreciate the reviewer’s careful attention to detail. We have removed the unnecessary single parenthesis after "mMIMO" as suggested.

Author action: The correction has been made in Section 3.2, paragraph one, on page 8 of the manuscript.

Reviewer 2, Concern 5: Define 〖(∙)〗^H, 〖(∙)〗^T and CN(0,1)

Author response: We thank the reviewer for pointing out the need for clarification. We have added definitions for the notations as requested.

Author action: The notation 〖(∙)〗^H and 〖(∙)〗^T has been defined in Section 3, page 6, and CN(0,1) has been defined in Section 7 of the manuscript.

Reviewer 2, Concern 6: Channel Model 3rd paragraph:

"The data set used for training includes real and imaginary channel gains, channel phase, spatial correlation, phase shift, correlation" - Is the second "correlation" a time related correlation?

Author response: We appreciate the reviewer’s question. To clarify, the second mention of "correlation" refers to spatial correlation between antenna elements, not time-domain correlation. Time-related correlation was not explicitly modeled in this dataset.

Author action: We have revised the sentence for clarity to specify that the second "correlation" refers to spatial correlation between antenna elements, and we have clarified that time-related correlation was not modeled.

Reviewer 2, Concern 7: The authors states: "Each AP server’s K UEs".

Does it mean that, if more than K UEs (let's say S) are placed in the scenario, an AP selection technique must be applied to assign K UEs, out of S, to each AP, or does it mean that the author modeled the system with a limited number of K UEs? Assigning a sub set of UEs to each AP, referred to as User-Centric Cell-Free, is known for providing a system scenario with reduced interference.

Author response: We appreciate the reviewer’s insightful comment. The statement "Each AP serves K UEs" refers to a system model where the total number of UEs assigned to each AP is fixed to K. We modeled the system with a limited number of K UEs per AP for simplicity and scalability in the simulation. No AP selection or dynamic user association was applied. If more than K UEs were placed, an AP selection technique could indeed be applied, but it was not part of this model.

Author action: We have clarified the statement and provided additional context

---

## [Decision Letter · Decision Letter 1]

13 Jun 2025

PONE-D-24-59811R1Spectral Efficiency and BER Analysis of RNN based Hybrid Precoding for Cell Free Massive MIMO Under Terahertz CommunicationPLOS ONE

Dear Dr. Abose,

Thank you for submitting your manuscript to PLOS ONE. After careful consideration, we feel that it has merit but does not fully meet PLOS ONE’s publication criteria as it currently stands. Therefore, we invite you to submit a revised version of the manuscript that addresses the points raised during the review process.

We look forward to receiving your revised manuscript.

Kind regards,

Nattapol Aunsri, Ph.D.

Academic Editor

PLOS ONE

Journal Requirements:

Reviewers' comments:

Reviewer's Responses to Questions

**Comments to the Author**

1. If the authors have adequately addressed your comments raised in a previous round of review and you feel that this manuscript is now acceptable for publication, you may indicate that here to bypass the “Comments to the Author” section, enter your conflict of interest statement in the “Confidential to Editor” section, and submit your "Accept" recommendation.

Reviewer #2: (No Response)

2. Is the manuscript technically sound, and do the data support the conclusions?

Reviewer #2: Yes

3. Has the statistical analysis been performed appropriately and rigorously? 

Reviewer #2: Yes

4. Have the authors made all data underlying the findings in their manuscript fully available?

Reviewer #2: Yes

5. Is the manuscript presented in an intelligible fashion and written in standard English?

Reviewer #2: Yes

6. Review Comments to the Author

Reviewer #2: (No Response)

7. PLOS authors have the option to publish the peer review history of their article (what does this mean? ). If published, this will include your full peer review and any attached files.

**Do you want your identity to be public for this peer review?** For information about this choice, including consent withdrawal, please see our Privacy Policy .

Reviewer #2: No

---

## [Author Response · Author response to Decision Letter 2]

24 Jun 2025

Original Article Title: “Spectral Efficiency and BER Analysis of RNN based Hybrid Precoding for Cell Free Massive MIMO Under Terahertz Communication”

To: PLOS ONE Editor

Re: Response to reviewers

Dear Editor,

Thank you for giving us the opportunity to revise the manuscript and address the reviewers’ comments.

We are uploading (a) our point-by-point response to the comments (below) (response to reviewers, under “Response to Reviewers”), (b) an updated manuscript with red highlighting indicating changes (as “Revised Manuscript with Track Changes”), and (c) a clean updated manuscript without highlights (“Manuscript”).

Best regards,

<Tadele A. Abose> et al.

Reviewer Concern 1: 〖SE〗_k=(〖 r〗k )/B (Verify if B was defined before)

Author response: We appreciate the reviewer’s attention to detail. The variable B in Equation (3) indeed denotes the system bandwidth. Although it is a standard notation, we acknowledge that it was not explicitly defined upon first use. We have now updated the manuscript to clearly define B as the system bandwidth to improve clarity.

Author action: We revised the text immediately following Equation (3) as in page 6.

Reviewer Concern 2: From Expression (1), one can expect that increasing the number of UEs K lowers SINR, and thus reduces SE. However, in Figure 5, SE appears to increase with K. Which SE metric is plotted: 〖SE〗_k, average SE, or total SE?

Author response: We thank the reviewer for the thoughtful observation regarding the trend in Figure 5. The curve in Figure 5 reflects the total spectral efficiency (SE), computed as:

〖SE〗_k=(〖 r〗_k )/B

As more users (UEs) are added to the system, while the individual or average spectral efficiency may decrease due to increased interference and resource contention, the total SE can increase since it sums the contributions from all users. This cumulative gain is consistent with CFMM systems, which scale network throughput as user density grows. We agree that explicitly stating this is necessary for clarity.

Author action: We revised the manuscript in the Results and Discussion section under Figure 5 on page 16.

---

## [Editor Report · Decision Letter 2]

2 Jul 2025

Spectral Efficiency and BER Analysis of RNN based Hybrid Precoding for Cell Free Massive MIMO Under Terahertz Communication

PONE-D-24-59811R2

Dear Dr. Abose,

We’re pleased to inform you that your manuscript has been judged scientifically suitable for publication and will be formally accepted for publication once it meets all outstanding technical requirements.

Kind regards,

Nattapol Aunsri, Ph.D.

Academic Editor

PLOS ONE
---

## [Editor Report · Acceptance letter]

PONE-D-24-59811R2

PLOS ONE

Dear Dr. Abose,

I'm pleased to inform you that your manuscript has been deemed suitable for publication in PLOS ONE. Congratulations! Your manuscript is now being handed over to our production team.

Kind regards,

on behalf of

Dr. Nattapol Aunsri

Academic Editor

PLOS ONE